# QoSRP: A Cross-Layer QoS Channel-Aware Routing Protocol for the Internet of Underwater Acoustic Sensor Networks

**DOI:** 10.3390/s19214762

**Published:** 2019-11-02

**Authors:** Muhammad Faheem, Rizwan Aslam Butt, Basit Raza, Hani Alquhayz, Muhammad Waqar Ashraf, Syed Bilal Shah, Md Asri Ngadi, Vehbi Cagri Gungor

**Affiliations:** 1Department of Computer Science, Universiti Teknologi Malaysia, Johor Bahru 81310, Malaysia; 2Department of Computer Engineering, Abdullah Gul University, Kayseri 38080, Turkey; cagri.gungor@agu.edu.tr; 3Department of Electronics Engineering, NED University of Engineering, Karachi 75270, Pakistan; rizwan.aslam@neduet.edu.pk; 4Department of Computer Science, COMSATS University Islamabad (CUI), Islamabad 45550, Pakistan; basit.raza@comsats.edu.pk; 5Department of Computer Science and Information, College of Science in Zulfi, Majmaah University, Al-Majmaah 11952, Saudi Arabia; h.alquhayz@mu.edu.sa; 6Department of Computer Engineering, Bahauddin Zakariya University, Multan 60800, Pakistan; wamuhammad4@live.utm.my; 7Department of Communication Engineering, Dalian University of Technology, Dalian 8435088, China; bilalshah@mail.dlut.edu.cn

**Keywords:** Internet of underwater things, channel aware, multichannel, acoustic sensor networks, underwater wireless sensor network

## Abstract

Quality of service (QoS)-aware data gathering in static-channel based underwater wireless sensor networks (UWSNs) is severely limited due to location and time-dependent acoustic channel communication characteristics. This paper proposes a novel cross-layer QoS-aware multichannel routing protocol called QoSRP for the internet of UWSNs-based time-critical marine monitoring applications. The proposed QoSRP scheme considers the unique characteristics of the acoustic communication in highly dynamic network topology during gathering and relaying events data towards the sink. The proposed QoSRP scheme during the time-critical events data-gathering process employs three basic mechanisms, namely underwater channel detection (UWCD), underwater channel assignment (UWCA) and underwater packets forwarding (UWPF). The UWCD mechanism finds the vacant channels with a high probability of detection and low probability of missed detection and false alarms. The UWCA scheme assigns high data rates channels to acoustic sensor nodes (ASNs) with longer idle probability in a robust manner. Lastly, the UWPF mechanism during conveying information avoids congestion, data path loops and balances the data traffic load in UWSNs. The QoSRP scheme is validated through extensive simulations conducted by NS2 and AquaSim 2.0 in underwater environments (UWEs). The simulation results reveal that the QoSRP protocol performs better compared to existing routing schemes in UWSNs.

## 1. Introduction

The ocean covers more than 70% of the Earth’s surface and it is vital for human life. It helps in driving weather and regulating Earth’s temperature, provides primary resources for humans and serves as a medium for commerce and transport. However, more than 95% of the volume of the ocean remains unexplored and, even more alarming, is unseen by human eyes due to lack of appropriate acoustic communication technologies for data collection from the ocean. However, the internet of things (IoT) technology is expected to make possible a new era in the vast ocean exploration with the aims to connect ubiquitous devices and facilities with different types of underwater networks to provide efficient and reliable services for all kinds of applications anytime and anywhere [1]. Thus, the use of Internet of Underwater Technology (IoUT) with underwater wireless sensor networks (UWSNs) technology can facilitate the discovery of unexplored marine resources. In fact, communication under water is more challenging due to the harsh nature of underwater environments (UWEs) [2]. Currently, most current underwater networks rely upon acoustic telemetry to communicate, but current acoustic modem technology is very limited by the low signal propagation speed in water (around $1.5\times 10^{3}\;$m/s as opposed to radio propagation of around $3\times 10^{8}$ m/s) and no commercial system can exceed a range × rate product of more than 40 km × 1 kbps [3]. However, the low bandwidth issue is not the only problem that is faced here but also dealing with many other problems such as high noise, path loss, multi-path signal propagation, Doppler spreading and high power consumption [4]. Putting this all together, we can see that acoustic networks have a large propagation latency, low available data transmission capacity, high bit error rate (BER) and dynamic topology structure in UWEs.

There are a few alternatives to acoustic telemetry, such as radio waves and optical communication that can be used [5]. The radio waves have been shown to be unviable for use in underwater networks as radio waves suffer from large signal attenuation and absorption unless extra low frequency (3–300 Hz) signals are used with long antenna size [6]. However, this comes at the expense of extremely high transmission power and low data rates depending upon the physical properties of the medium characteristics on the other hand, the optical modems can be used in conjunction with acoustic telemetry to achieve higher data rates associated and a lower error rate than acoustic telemetry. However, the optical signal can only propagate short distances (e.g., 2−8 m with special lenses) [7]. This solution becomes useful when the acoustic sensor nodes (ASNs) are close to each other and there are no obstructions in the water, such as oil, fish, rocks, etc. The short range means that optical signals are not useful due to heavy scattering and are restricted to short-range-line-of-sight issues in acoustic networks for time-critical events monitoring purposes. Thus, these approaches are expensive, not scalable and intolerant to faults.

To this end, underwater acoustic communication seems to be the best option and, therefore, underwater acoustic sensor networks (UASNs) have been attracting increasing attention from scientific and industrial communities [8]. The ASNs in UASNs have sensing, processing and communication capabilities, thus they are emerging as a fundamental technology for IoUT. The use of underwater sensor nodes enabled with wireless communication capabilities have the potentials to realize real-time underwater monitoring and actuation applications, with an on-line system reconfiguration and failure detection capabilities [9]. Therefore, this technology is making possible a new era in variety of ocean monitoring and exploration applications, such as monitoring natural disasters, mine recognition, navigation assistance, underwater pollution, study of marine life and tactic surveillance applications [10]. These applications will help in filling the gap of our knowledge regarding the ocean and aquatic environments in general. Nevertheless, deployment of UASNs due to their acoustic modem transceiver and the appropriate hardware for protecting the circuitry is expensive and limited to experimental settings. The ASNs deployment missions can take several days since sensors might be attached to docks, anchored buoys, sea floors, underwater autonomous vehicles (UAVs), low-power gliders, or underpowered drifters, depending on the desired network architecture [5].

On the other hand, efficient data collection using ASNs is challenging due to the various aforementioned factors in the UWSNs. Often, underwater ASNs are equipped with acoustic modems to wirelessly communicate with each other. Underwater communication links are affected due to location and time-dependent acoustic channel communication characteristics. Moreover, temporary connectivity loss can occur due to shadow zones. Therefore, the wireless link between neighboring ASNs may perform poorly or even be down at any given moment, which will increase the packet retransmissions as an attempt to deliver data packets result in more packet collisions, delay and energy consumption in UWSNs. Thus, excessive rerouting due to poor link quality consumes nodes residual energy, which can disrupt underwater monitoring and exploration missions prematurely [11]. To address these issues, a number of data collection protocols have been proposed in the literature (see Section 2). The key aim of these routing schemes is to mitigate the impacts of the underwater environments and enhance the poor quality of underwater acoustic physical links by taking advantage of the multicast and broadcast nature of the wireless transmission medium. However, each approach has disadvantages that, when combined, could diminish significantly the performance of UWSNs. Some of their common drawbacks, e.g., poor data reliability and channel capacity, communication void region problem, high latency, and redundant data packet transmission are accented in acoustic communication, which severely diminishes UWSNs performance. Therefore, a highly reliable routing architecture is essential for UWSNs.

In the rest of the paper, Section 2 discusses the existing routing protocols in UWSNs. In Section 3, we briefly describe the proposed quality of service routing protocol (QoSRP). In Section 4, we discuss the acoustic channel and energy consumption models used in the simulation studies. In Section 4, we also provide simulation results and analyze the performance of QoSRP against existing routing protocols in UWEs. Finally, in Section 5, we conclude the paper with future work.

## 2. Existing Studies and Challenges in Underwater Wireless Sensor Networks (UWSNs)

In the past few years, there has been intensive research in designing routing schemes for UWSNs. In this respect, this section summarizes up to date existing studies on the developments and status of routing schemes in UWSNs. The authors in [12] propose a depth based routing protocol to minimize the end-to-end latency and energy consumption in UWSNs. The designed scheme considers an optimal weight function to compute the speed of receiving signals and transmission loss during relaying data over shortest paths in low-depth regions. The proposed scheme minimizes the latency and energy consumption at the expense of low network throughput and packet delivery ratio in UWSNs. Similarly, a depth-based anycast geographic and opportunistic routing protocol is proposed in [13] for UWSNs. The designed scheme employs a recovery mode procedure based on the ASNs depth adjustment during relaying data packets from ASNs in void regions to the sea surface sink. The research in [14] balances the energy consumption load and avoids the void regions because of considering the normalized depth variance of the relay ASNs in the network. Also, in [15] an idea of pressure routing is discussed to provide reliable packets transmission of the events to any surface sonobuoy. The proposed scheme selects the subset of relay ASNs based on the pressure levels to route data at low interference towards the surface sonobuoy. The work in [2] proposes a location-free pressure routing protocol that considers three basic parameters such as depth, residual energy and link quality of the relay ASNs. Both above routing schemes achieve low latency, energy consumption and high packets delivery with the expense of poor synchronization and corrupted data packets due to detouring forwarding in void regions in UWSNs.

To solve the issues of above-mentioned depth-based routing schemes, the work in [16] presents a novel concept of vector-based opportunistic routing in which the smallest hop counts are selected towards the sink during conveying data packets. The developed scheme due to its highly stable routing architecture performs well in achieving low latency, energy consumption, and high packet delivery rates in UWSNs. The work in [17] discusses a packets forwarding mechanism for marine monitoring applications. The proposed scheme employs various transmitting power levels and the residual energy of down-stream relay ASNs in the information relaying process. The developed protocol prolonged the network lifetime due to its low energy consumption, however, it faces the issues of high latency and low data delivery rates in UWSNs. Similarly, the study in [18] presents a tree-based routing protocol for reliable data gathering in UWSNs. The entire working mechanism is divided into two phases, namely, dynamic tree constructions and information gathering using an autonomous underwater vehicle. Initially, a set of gateway nodes along the shortest path trees is defined to limit the association count of neighboring ASNs. Then, an autonomous underwater vehicle is used to collect data from gateways to prevent data loss in the network. The proposed protocol achieves low latency performance and increases the packets delivery rates and throughput, but it faces the issues of communication overheads in UWSNs.

The protocol proposed in [19] exploits the link quality of relay ASNs by considering the cross-layer design paradigm during forwarding packets around connectivity voids and shadow zones in UWSNs. Similarly, the research in [20] discusses a cross-layer location-free single-copy protocol for reliable data gathering in UWSNs. The developed scheme due to its dynamic controls transmit power and channel frequency mechanisms avoids overhead introduced by redundant packets, balance energy consumption and increases the packet delivery rate in UWSNs. Also, the study in [21] presents a cross-layer cooperative routing protocol in which the ASNs selects the forwarding relay ASN based on their link quality indicators towards the destination. These schemes perform well in terms of low energy consumption, latency and high packet delivery ratio in UWSNs. However, the first scheme faces the issue of high congestion near to the sink than others while the main problem of the second scheme is invalid data packets due to path loops in the network. On the other hand, the third routing protocol faces excessive overhead issues due to periodically updating the neighboring ASNs in UWSNs. To solve some of the issues faces by above schemes, the authors in [22] discuss a novel deep Q-network based idea to make globally optimal routing decisions. The designed scheme to reduce network overhead employs a hybrid unicast and broadcast communication mechanisms to maintain network topology changes and making new routing decisions when the current data paths become unavailable.

The protocol in [23] divides entire grid routing architecture into many small cubes indicated as clusters to maintain the reliability of data transmission. In each cluster, a cluster leader is appointed based on the location and remaining energy information to maintain the reliability of data transmission. Similarly, the study in [24] provides a bio-inspired distributed spectral clustering routing protocol that exploits link quality by considering signal to noise ratio (SNR) values between relay ASNs to provide highly stable clustering and routing architecture in the network. The research in [10,25,26] propose bio-inspired dynamic cluster-based routing protocols to provide reliable data transmission by considering the number of hop counts and the confidence level of the relay ASNs in UWSNs. Also, in [27], a hybrid data-collection routing protocol is proposed in which the entire working mechanism is divided into the upper layer and lower layer to perform reliable data gathering in UWSNs. The simulation facts show that the proposed schemes achieve low latency, energy consumption and higher packet delivery rates in the network. However, the first scheme faces high routing table management cost, and the second scheme compared to other schemes faces the corrupted data packets due to poor link quality among cluster heads and cluster heads rotating issues in highly dense UWSNs. The study in [28] presents a disjoint multipath disruption-tolerant packets forwarding mechanism in which multiple routing paths by employing the hue, saturation and value colour space that are constructed to greedily convey packets to the sea surface sink. The developed protocol improves the packets delivery rates and reduces the latency, but it faces the issue of low throughput and poor loading balancing in UWSNs.

Han et al. [29] propose an asymmetric link-based reverse routing in which a directional beam width mechanism is used to analyze the link quality between relay ASNs. The proposed protocol eliminates void issues and improves network performances in terms of latency and packets delivery rates. However, this faces the issue of high communication overheads and packets collision due to periodically updating the status of the links in UWSNs. Similarly, the protocol proposed in [30] uses multi-layered architecture to discover a set of feasible relay ASNs during conveying data towards the sink. The proposed protocol achieves better performance in low energy consumption and packet delivery rates with low latency, but it faces the issues of poor synchronization and invalid data packets in UWSNs. To address some of the issues faced by the above schemes, the protocol in [31] overcomes packets loss by considering the forwarder ASNs connectivity issues along a route towards the sink. The performance of the proposed scheme is observed better in low latency, energy consumption and packets error rates in UWSNs. Table 1 shows the comparison between different routing schemes in UWSNs.

The main purpose of the aforementioned routing protocols is to provide reliable data delivery at a low cost in UWSNs. These static-channel based schemes significantly help in designing and development of advanced routing solutions for UWSNs. However, they suffer from major disadvantages. First, most of the schemes employ probabilistic values to estimate the link quality between ASNs in highly dynamic UWEs. Thus, the poor link quality due to the false estimation of the channel is facing severe reliability issues lead to excessive rerouting during monitoring time-critical events in UWSNs. The new routes finding or repairing broken links require a significant amount of control message overheads, which consumes the sensor’s energy and increases the chance of packets collision and brings latency issues in the network. On the other hand, the interference issues further increase the corrupted data packets in UWSNs. Second, during conveying packets these schemes consider shortest paths routing with excessive hop counts, which may balance the network residual energy; however, it consumes more ASNs energy and increases the probability of invalid data packets because of data path loops in UWSNs. In addition, the shortest path routing result in high routing table management cost and congestion problems because of quickly draining the residual energy frequently used by ASNs nearer to the sink. Third, most of these routing protocols are stuck in void regions and instead of finding an alternative route just discards the packets and thus subjected to packet loss in UWSNs. Fourth, the existing static-channel based routing solutions due to lack of dynamic channel adaptation cannot mitigate the inference effects in order to provide high packet delivery rates and network throughput with low latency and corrupted packets in UWSNs.

These facts motivate researchers to propose a novel cross-layer QoS-aware multichannel routing protocol called QoSRP for the internet of UWSNs to mitigate the effects of the underwater environments and improve the overall data collection in UWSNs. The entire routing problem has been modelled using mixed integer linear programming (MILP) in UWSNs. The major contributions of our proposed protocols are as follows:We propose an underwater acoustic channel detection mechanism to find the vacant channels with a high probability of detection and low probability of missed detection and false alarms.An underwater acoustic channel assignment mechanism is proposed to assign high data rates channels to acoustic sensor nodes with longer idle probability in a robust manner.A hybrid underwater routing mechanism is proposed to convey collected data to the sink. The proposed mechanism while conveying information avoids congestion, data path loops and balances the energy consumption load of UWSNs.The performance of the QoSRP protocol against existing routing schemes is validated through extensive simulations conducted by NS2 and AquaSim 2.0 in the UWSNs.

The later sections explain the working mechanism of our proposed routing protocol in UWSNs.

## 3. Proposed Quality of Service Routing Protocol (QoSRP) in UWSNs

The protocol design details are given in the following sections.

### 3.1. Network Model

The network model used in the design, development, simulating and testing the QoSRP scheme is illustrated in Figure 1. The proposed architecture consists of a sea surface sink, base station (BS) and ASNs. The randomly deployed ASNs in a geographic area of interest over the ocean bottom are equipped with the main functions of sensing, sampling, and acoustic transmitters. These location-aware ASNs (computed by using self-localization scheme in [32]) deployed for continuous oceanographic data collection are equipped with omnidirectional acoustic transceivers have identical communication range and asymmetric communication links. In UWEs, the ASNs due to constraints of limited residual energy and short communication range transmit packets to the sink in a multiple hops manner. Thus, the ASNs die once the energy runs out. In addition, the ASNs employing different channels on multiple paths with different lengths have different propagation period, the angles of arrival (AoA) and the angle of departure (AoD) in UWEs. In addition, the ASNs move passively with water currents in vertical and horizontal directions with a velocity of 0 to 0.8 m/min and 0 to 1 m/min, respectively. This extremely slow movement in both upward and parallel directions is assumed negligible in the UWEs. On the other hand, the sink floats on the sea surface and is equipped with both acoustic modem and radio modem aiming to gather data from ASNs through acoustic signals and forward the gathered data using radio signals to the offshore BS and then the remote user/s for monitoring purposes. The sea surface sink is embedded with a global positioning system (GPS) and periodically updates the BS about its location information by disseminating periodic beaconing in the network. The energy of both sea surface sink and BS is assumed to be unlimited since they can be charged by suitable green energy resources, such as solar energy, etc. The technology such as carrier sense multiple access (CSMA) is assumed to avoid data packets collision during transmission and reception in UWSNs. Finally, we assume that the remote user/s can configure, control and monitor ASNs by connecting to the BS through one of the highly stable communication technology, such as satellite or cellular. The following section explains the working procedure of QoSRP protocol in detail.

### 3.2. Underwater Acoustic Channel Detection (UWCD) Algorithm

Recently, the spectrum shortage problem aggravates in an apparent manner because of increasing demands for higher data rates at low frequencies for various UWSN-based applications. The principle of dynamically exploiting the local vacant spectrum bands of the primary users during their silence periods by secondary users is among the best-proposed solutions for this problem. Therefore, the multichannel is proposed as a prominent technology to implement spectrum access for its autonomous, agility, and ability to detect the primary user’s (PU) signal. In multichannel communications, the spectrum sensing is a key element in the sense that it demonstrates how signals of the PUs are identified, sampled, and processed for detecting spectrum holes [33]. The main tasks accomplished by spectrum sensing, such as spectrum monitoring, spectrum analysis, and spectrum decision while the multichannel cycle consists of four core operations, including the above three and data transmission. Thus, the key aim of the spectrum sensing is to search spectrum holes for secondary users by detecting the primary signals in the network. To sense the accessibility of certain portions of the frequency band, the most effective way is to identify the primary users that are active within the range of a secondary user. However, the direct measurement between a transmitter and receiver for a channel is difficult and brings several new challenges for the secondary users in the UWEs. Generally, there exist few signal detection techniques like matched filter detection (MaD), wavelet detection (WeD), cyclostationary detection (CyD) and energy detection (EnD) [34], etc., that can be employed during sensing the spectrum to enhance the detection probability in the UWEs.

In the MaD technique, the SU have the prior knowledge of the PU signals and a matching filter is applied, which amplifies the SNR of the received acoustic frequency samples. This mechanism, due to coherent detection, achieves a high processing gain and thus reduces the channel detection latency. However, each primary user class requires a dedicated sensing receiver, which is expensive and consumes significant ASNs battery power in UWSNs. Moreover, the channel detection performance of the matched filter can be severely limited because the primary user (PU) signals information is hardly available at the $PU{}^{\prime }𝓈$ in UWEs. By contrast, in the WeD approach, the given signal is first split into various frequency components and later every component by matching the resolutions to its scales is studied. The wavelet detection basically employs cosines and sines as key functions [35]. The use of irregularly shaped wavelets in the wavelet transforms as basic functions to identify the local features and sharp changes. This approach offers flexibility in dynamic channel adaptation and low implementation cost. However, the high sampling rates for characterizing the large bandwidth is one of the most critical challenges while implementing the wavelet approach. In the CyD technique, the presence of primary users can be identified even at very low signal-to-noise values by observing the periodicity of the collected $PU{}^{\prime }𝓈$ signals in UWEs. In the cyclostationary detection to obtain the periodicity of the primary signals, the modulated signals are usually combined with hopping sequences, spreading code, pulse trains, and cyclic prefixes or sinusoidal carriers. These modulated signals due to exhibiting the characteristics of periodic statistics and spectral correlation are characterized as cyclostationary [36].

Typically, these features in the signal format are introduced intentionally, which enables a receiver to exploit parameter estimation like the direction of arrival, pulse timing, or carrier phase. This detection technique can improve the sensing detection, however, it requires prior knowledge of the primary signal characteristics. In addition, the high computational complexity and significantly extensive sensing delays are the other shortcomings of this approach. The EnD is the widely used channel detection method, which does not require any previous knowledge of the PU signals. This method identifies the primary user’s signals based on the sensed energy, where the received signal strength indicator or acoustic-frequency energy is computed to find whether the channel is free or not. Initially, an input signal to obtain the required bandwidth is filtered through a band pass filter and then the obtained signal is squared and combined over the observation interval. Lastly, to find the presence of a primary user signal a predetermined threshold is compared with the output of the integrator. Generally, fast Fourier transform-based techniques are used to analyze the spectral in the digital domain. In particular, in a specific time window, the received signal is sampled and passes through fast Fourier transform equipment to obtain the power spectrum. Then, the power spectrum peak after windowing is detected in the frequency domain. This technique provides better channel detection in underwater since it does not need any previous information about the $PU{}^{\prime }𝓈$ signals. However, the high noise, interference, fading, and multipath effects increase the probability of sensing errors in terms of false alarms (FA𝓈) or miss-detection (M𝒾D), which leads to poor channel detection performance in UWSNs. Hence, it is highly desired to reduce the sensing errors during channel detection to enhance the usage level of the vacant channel and to minimize the collision probability with PU transmissions.

In this respect, the detection threshold that precisely detects the presence of primary signals plays an important role in the multichannel acoustic sensor networks (MCASNs). The less error in computing the SNR value decreases the probability of FA𝓈 in UWEs. However, there exists a limit for the noise level. The increase in noise level above this limit increases the probability of FA𝓈 while it reduces with the decrease in noise below this threshold level. In the proposed scheme, the channel detection threshold level is determined by considering the various underwater environment parameters and set to a minimum level of the transmitted signal power value in the underwater. This detection sensitivity identifies the received SNR ratio at the secondary user (SU) with a satisfied probability of detection conditions in MCASNs. Thus, the optimal channel detection threshold significantly reduces the sensing errors, enhance spectrum utilization and provides enough protection to PU transmissions in the MCASNs. In the designed scheme, a simple and effective energy detection mechanism appropriately detects the existence of the PU signal based on its received energy by comparing it to a detection threshold. Herein, we employ an energy-based signal detection (E) method revealed in [37] to identify idle spectrums in UWSNs. The key objective function of the EnD mechanism $\left(\phi_{UWCD\left(1\right)}\right)$ of the proposed algorithm is numerically indicated as:∀ 𝒾= {1,2,…,𝓃};∀ 𝒿= {1,2,…,𝓂};∀ 𝓀= {1,2,…,ℓ};

(1)$$\phi_{UWCD\left(1\right)}=\overset{\left|n\right|}{\underset{i=1}{\sum }}\underset{i}{\max}\rho_{DℯT}+\underset{i}{\min}\rho_{FA𝓈}+\;\underset{i}{\min}\rho_{M𝒾D}\;$$

The key aims of the objective function is to maximize the probability of detection $\left(\rho_{DℯT}\right)$ and minimize the probability of FA𝓈
$\left(\rho_{FA𝓈}\right)$ and missed-detection $\left(\rho_{M𝒾D}\right)$ in UWSNs. Consequently, the test statics of energy detection $\left(T_{ΕD}\right)$ of the received signal at the $SU_{𝒾}$ can be numerically written as:(2)$$T_{ΕnD}=\overset{L_{𝓈}}{\underset{𝓃=1}{\sum }}{\left|ΕS_{L}\left[𝓃\right]\right|}^{2}\;$$
subject to:(2a)$$S_{L}\left[𝓃\right]\leq \tau_{\varsigma }\;;$$

(2b)$$0<M_{sg\left(i\right)}\leq Msg_{sg\left(j\right)}\leq 1;$$

(2c)𝓃 >1≤k;

(2d)$$ΕS_{L}\left[𝓃\right]\geq {Ε}_{\varsigma };$$

(2e)$$\rho_{ℯ}\leq \;\rho_{ℯ}\left(\epsilon \right)<1;$$

(2f)$$\tau_{\varsigma }>0\;1\leq \varsigma \leq T\;\forall 𝓃\in PU_{𝒿};$$

(2g)$${Ε}_{\varsigma }\geq 0\;1\leq \varsigma \leq E\;\forall E\left[𝓃\right]\in \;PU_{𝒿};$$

(2h)$$\rho_{ℯ}\geq 0\;1\leq e\leq E;$$

In Equation (2), $ΕS_{L}$ and $S_{L}\left[𝓃\right]$ are, the received signal energy and the 𝓃-sample of the $S_{L}$ while $L_{𝓈}$ is the sum of the length of 𝓃 samples over an interval which helps to obtain a level of performance under certain SNR conditions in the network. The number of samples collected by individual SU depends upon the sensing time and the longer sensing delay within in predefined time intervals leads to better detection performance of the primary signals. However, the longer sensing time results in less available time for packets transmission, which minimizes the throughput of the SU𝓈 is constrained by Equation (2a). Moreover, due to additional sensing time the cooperation overhead with the increasing number of cooperating users’ increases, which lead to a huge volume of information that need to process by the requested SU to make a local decision is constrained by Equation (2b). Therefore, the number of samples of the received signal energy of $L_{𝓈}\left(n\right)$ such that 𝓃
>1 using the central limit theorem over an interval are considered to obtain some level of performances under certain SNR conditions is constrained by Equations (2c) and (2b) in the network. Therefore, it is indicated as one of the basic functions of the SNR in the UWEs. The transmission of PU signals is a random process, which follows an independent identically distributed pattern with mean zero and variance $\sigma_{𝓈}^{2}$. In the underwater, the noise $\left({𝓃}_{oise}\right)$ is a real-valued Gaussian variable that considers various underwater environment parameters with zero mean and variance $\sigma_{𝓋}^{2}$ is constrained by Equation (2e) within a certain error probability. Constraints in Equations (2f, h) satisfy the limits in the network. To this end, the SNR in the underwater at the 𝒾th secondary user $SU_{𝒾}$ is computed as $SNR\left(SU_{𝒾}\right)={\left|C_{ℊ}\right|}^{2}\sigma_{𝓈}^{2}/\sigma_{𝓋}^{2}$. The primary user signal and noise are assumed to be independent and therefore a binary hypothesis testing problem of received energy signals at the SU can be formulated as:(3)$$T_{ΕnD}{\left(SU_{𝒾}\right)}_{S_{L}}=\overset{𝓂}{\underset{𝒾=1}{\sum }}{\left|{𝓃}_{o\left(𝒾\right)}\right|}^{2}\in \;H_{0}\;$$
(4)$$T_{ΕnD}{\left(SU_{𝒾}\right)}_{S_{L}}=\overset{𝓂}{\underset{𝒾=1}{\sum }}{\left|C_{ℊ\left(𝒾\right)}S_{L\left(𝒾\right)}+{𝓃}_{o\left(𝒾\right)}\right|}^{2}\in \;H_{1}\;$$
in which 𝓂, $C_{ℊ\left(𝒾\right)}$, ${𝓃}_{oise\left(𝒾\right)}$ are the time-bandwidth product, the channel gain between PU and SU, and noise in the ${𝒾}^{𝓉𝒽}$ time slot, respectively. The received energy at each $SU_{𝒾}$ is compared to the predefined detection threshold (σ) to reach a decision about the presence or absence of $PU_{𝒿}$. In locally sensing mechanism each SU decides whether the 𝒾th channel is available ($\tau_{\mathrm{off}})$ or occupied ($\tau_{on}$) by using a predefined threshold in the network. Thus, the hypothesis at each secondary user is saying that the PU is active only if the received signal energy is greater than the defined limit and idle when it is less than the defined threshold value, which can be numerically indicated as:(5)$$T_{ΕnD}{\left(SU_{𝒾}\right)}_{S_{L}}=\left[H_{0}\in ΕS_{L}>\sigma \;|H_{1}\in ΕS_{L}<\sigma \right]\in \tau_{s}$$
subject to:(5a)$$T_{ΕnD}{\left(SU_{𝒾}\right)}_{S_{L}}=\;\tau_{\mathrm{on}}|\tau_{\mathrm{off}}\in C_{𝒾}\subseteq C_{𝓃}$$
(5b)$$H_{1}\left(C_{i}\right)\in C_{n}=1\left(\tau_{\mathrm{on}}\right)$$
(5c)$$H_{0}\left(C_{i}\right)\in C_{n}=0\left(\tau_{\mathrm{off}}\right)$$
(5d)$$PU_{𝒿}\left(C_{i}\right)\in C_{n}\neq \tau_{\mathrm{on}}$$
(5e)$$C_{i}\in H_{1}|H_{0}\;\leq 1$$
(5f)$$SU_{𝒾}\left(C_{i}\right)\in C_{n}\left(R_{𝒾}\right)=\tau -\tau_{s}$$
(5g)$$SU_{𝒾}\notin C_{i}\left(R_{𝒾}\right)=\tau$$
(5h)$$SU_{𝒾}{\left(C_{i}\right)}_{busy}\in C_{n}\left(R_{𝒾}\right)\neq \tau_{s}$$
in which σ is the threshold value of energy is used to compare is the local spectrum sensing decision of a $SU_{𝒾}$,$\;H_{1}$ denotes the 𝒾th channel is not available because of the $PU_{𝒾}$ activity with probability $\rho_{\left(PU_{𝒾}\right)}$ in the sensing time $\tau_{s}$ is constraint by Equation (5a). Equations (5b) and (5c) constraints guarantee that the PU is active and inactive for a particular channel $C_{i}$ in a time interval in the network. Equation (5d) constraints state that a channel belongs to the $PU_{𝒿}$ is available to $SU_{𝒾}$ for the time $\tau_{s}$, while the constaints in Equation (5e) satisfy the satement specified in Equation (5d). Constraints in Equation (5f) verify that the $SU_{𝒾}\;$senses a channel $C_{i}$ in the region $R_{𝒾}$ and holds the channel for a specific time once it is found free in the network. Constraints in Equation (5f) explain that the $SU_{𝒾}$ must not hold the channel for the entire time since the $PU_{𝒿}$ can reclaim the channel any time in the network. Constraints in Equation (5h) state that the $SU_{𝒾}\;$senses a channel $C_{i}$ in a region $R_{𝒾}$ and will not be sensed the same channel immediately once it is found busy in the network. On the other hand, $H_{0}$ illustrates that the 𝒾th spectrum is available with probability$\;\rho_{𝒾𝒹ℯ𝒶\ell }$, i.e., $\rho_{\left(PU_{𝒾}\right)}+\rho_{𝒾𝒹ℯ𝒶\ell }=1$ in the sensing time $\tau_{s}$. The performance of locally detecting spectrum by each SU is computed by the probability of detection when a $PU_{𝒾}$ is idle and the probability of FA𝓈 when a $PU_{𝒾}$ is active and can be numerically expressed as

(6)$$\rho_{DℯT}=\rho \left(H_{1}|H_{1}\right)\in \rho \left(ΕS_{L}>\sigma |H_{1}\right)$$

(7)$$\rho_{M𝒾D}=1-\rho_{DℯT}\left(ΕS_{L}<\sigma |H_{1}\right)$$

(8)$$\rho_{FA𝓈}=\rho \left(H_{1}|H_{0}\right)\in \rho \left(ΕS_{L}>\sigma |H_{0}\right)$$

Equations (6), (7) and (8) show the DℯT probability when SU determines $H_{1}$ when $H_{1}$ exits, the M𝒾D probability when SU decides $H_{0}$ but $H_{1}$ exists, and the FA𝓈 probability when SU decides $H_{1}$ but $H_{0}$ exists, respectively. In addition, Equations (7) and (8) clearly indicate that SU provides reliable protection to PU when the detection probability is high while the SU loses spectrum access opportunities when the false alarm probability is high in the network. Some common numeric terms used throughout the paper are given in Table 2.

#### 3.2.1. Decision Rules for the Fused Information in Distributed Cooperative Sensing (DCoS)

Distributed cooperative spectrum sensing consists of main two types of data fusion, namely hard decision and soft decision mechanisms in the MCASNs. The soft decision mechanism due to heavily sharing test statistics faces excessive cooperation overheads compared to the hard decision mechanism, which consumes the sensor’s energy and increases the probability of channel assignment delay in MCASNs. On the other hand, the hard decision mechanism performs best when the channel state information between the PU𝓈 and SU𝓈 varies in time and location in UWEs. The hard decision mechanism after considering the individual SU𝓈 local decisions conveniently applies the linear fusion rules such as AND, OR, and K-out-of-M to obtain a reliable cooperative decision. In the AND fusion rule, the fusion center declares $H_{1}$ only if all independent SU𝓈 decide on $H_{1}$ and declares $H_{1}$ for the OR rule only if any of independent SU𝓈 decides on $H_{1}$ in the cooperative decision. On the contrary, the fusion center in K-out-of-M rule declares the ultimate decision that there is a $PU{}^{\prime }𝓈$ transmission only if at least K secondary users out of M selected local detectors decide about the presence of a $PU{}^{\prime }𝓈$ signal. In the proposed scheme, the probability of FA𝓈, M𝒾D and DℯT using K-out-of-M rule can be written as:(9)$$\rho_{DℯT}=\overset{\left|𝓃\right|}{\underset{\ell =𝓀=1}{\sum }}\overset{\left|𝓂\right|}{\underset{𝒿=1}{\sum }}\rho_{DℯT}{\left(SU_{𝒿}\right)}^{\ell }\;{\left(1-\rho_{DℯT\left(SU_{𝒿}\right)}\right)}^{\left|𝓂-\ell \right|}$$

(10)$$\rho_{M𝒾D}=1-\overset{\left|𝓂\right|}{\underset{𝒿=1}{\sum }}\left[\rho_{DℯT}\left(1-\rho_{ℯ}\right)+\left(1-\rho_{ℯ\left(SU_{\;𝒿}\right)}\right)+\rho_{ℯ\left(SU_{\;𝒿}\right)}\right]$$

(11)$$\rho_{FA𝓈}=1-\overset{\left|𝓂\right|}{\underset{𝒿=1}{\sum }}\left[\left(1-\rho_{FA𝓈}\right)\left(1-\rho_{ℯ}\right)SU_{\;𝒿}+\rho_{FA𝓈\left(SU_{\;𝒿}\right)},\rho_{ℯ\left(SU_{\;𝒿}\right)}\right]$$

Equations (9), (10) and (11) illustrate the probability of DℯT when all 𝓂 secondary users sensed the existence of a $PU_{𝒾}$ such that $SU_{𝒿}>SU_{i}$ determine $H_{1}$ when $H_{1}$ exits, the M𝒾D probability when the secondary users $SU_{𝒿}>SU_{i}$ decides $H_{0}$ but $H_{1}$ exists, and the FA𝓈 probability when the secondary users $SU_{𝒿}>SU_{i}$ decide $H_{0}$ but $H_{1}$ exists, respectively.

#### 3.2.2. Distributed Cooperative Sensing (DCoS)

In the UWCD scheme, the local channel detection made by the individual SU for the desired capacity-aware channel is preferred. However, the high underwater noise, fading, and interferences have destructive effects caused by imperfect reporting channels on the wireless link quality leading to poor cooperative spectrum sensing in the network. This problem becomes more severe between the sender and the receiver secondary users due to an increasing probability of errors over the reporting channels of the transmitted signal in the network. Therefore, finding a channel with high data rates become more difficult in local spectrum sensing in UWEs. To this end, the distributed cooperative sensing (DCoS) provides better results in highly dynamic UWEs. The SU𝓈 in the proposed DCoS architecture performs the channel sensing tasks on demand only if the required capacity-aware channel is not found locally by the SU. To do so, the SU that requires the channel information sends multicast request messages to its neighboring ASNs. The request message includes the required channel information, identity, and location information in the network. Upon receiving the request message, the neighboring SU𝓈 start to sense the required channels and share their channel detection statistics by exploiting the spatial diversity in the observations. Thus, each SU based on its local sensing observation exchanges decision on the existence or non-existence of the primary signals in the UWSNs.

The receiver SU combines the received sensing information with its own information and after some necessary calculations makes an ultimate decision whether the PU is active or not in the network. This decision information is disseminated to each sender SU𝓈 in the network. This combined decision mechanism significantly improves the performance of channel detection and relaxed sensitivity requirements for the SU in the network. Thus, each receiver SU acts as a fusion center as shown in Figure 2. In Figure 2, a $SU_{𝒾}$ that requires a specific channel/s information sends a request message to its neighboring nodes. The neighboring SU𝓈 after successfully receiving the request message start to observe the channel activity in the network. Then each SU based on local observation decides the presence or absence of the primary signal and forwards its final decision to the request SU by considering the CSMA mechanism in the MCASNs. The selected SU𝓈 perform channel detection in a group manner in a way such that if required then more than one channel are sensed in parallel in every sensing interval as shown in Figure 2. During the information exchange process, an acknowledgment message is forwarded by the receiving node, which guarantees that the information has been received successfully.

This mechanism significantly reduces the cooperation overhead in terms of latency for searching an appropriate spectrum since multiple channels are sensed in a single sensing interval. Moreover, it also significantly minimizes the network implementation cost by eliminating the need for centralized intelligent devices in the UWSNs. In addition, the fusion mechanism improves the robustness of the decision-making process at the fusion node on the presence or absence of PU𝓈 in UWSNs. The cooperative decentralized parallel channel sensing architecture of our proposed scheme is indicated in Figure 3. It illustrates that different SU𝓈 observe the designated channel/s and each of them makes an individual decision and forwards its one-bit decision to the fusion center in a parallel manner in UWSNs.

#### 3.2.3. User Selection in Cooperative Mechanism

In UWSNs, the fusion center cooperating with all users does not essentially attain the optimal performance since the channel fluctuates over time and location due to the harsh nature of the UWEs. This might lead to assigning a poor-quality channel to the SU which degrades the performance of the network. Therefore, the selection of SU𝓈 for obtaining the decisions for specific primary signals plays an important role since it can be used to increase the sensing performance at the fusion center. Thus, the most appropriate SU𝓈 among the others which have better DℯT probability by considering a given FA𝓈 probability in a region must be given the opportunity to participate in the DCoS. In the UWCD scheme, the appropriate cooperating SU𝓈 raise the probability of DℯT and minimize the total error probability for the SU, and thus improve the total throughput of the network. In addition, it provides sufficient protection to the PU𝓈 in terms of interference in the network. A set of secondary users selected $S_{e}\left(SU_{𝒾}\right)$ from the rest of the secondary users $SU_{𝒿}$ in the decision-making procedure can be numerically shown as:(12)$$S_{e}\left(SU_{𝒾}\right)=\overset{𝓃}{\underset{𝒾=1}{\sum }}\overset{𝓀}{\underset{𝒿=1}{\sum }}SU_{𝒿}-SU_{𝒾}\;$$
subject to:(12a)$$SU_{𝒾}\subseteq SU_{𝒿}\in R_{𝒾}\left(C_{i}\right)<R_{𝓃}\;$$

(12b)$$SU_{𝒾}\in I_{𝓫ℯ𝓈𝓉}\left(H_{1}|H_{0}\right)≅1$$

(12c)$$\rho_{ℯ}SU_{𝒾}\left(C_{i}\right)<\delta$$

(12d)$$SU_{𝓀}\in I_{𝓅οο𝓇}\left(H_{1}|H_{0}\right)=0$$

(12e)$$SU_{𝒾}\left(H_{1}|H_{0}\right)\leq \tau_{s}\;<\tau$$

Constraints in (12a) state that the selected secondary users are less than the total number of secondary users involved in the sensing process for the distinct channels in a region in the network. Constraints in (12b) guarantee that the selected secondary users perform the best $\left(I_{𝓫ℯ𝓈𝓉}\right)$ in terms of channel detections with low error probability (δ) is satisfied by Equation (12c). Constraints in (12d) state that the secondary users with poor detection probability $\left(I_{𝓅οο𝓇}\right)$ are removed from the channel detection group in the network. Constraints in (12e) make it sure that the selected secondary users detect the channel in predefined time intervals $\left(\tau_{s}\right)$.

#### 3.2.4. Sensing Overhead at the Fusion Center

The local sensing and data reporting to the fusion center spend a notable amount of SU𝓈 energy in the MCASNs. In this respect, the limited sensing information by following certain criteria or constraints, which avoids unnecessary or uninformative data reporting can significantly improve energy efficiency in cooperative detection in the MCASNs. Realizing the facts, the proposed scheme employs a few sensing bits from each neighboring so that SU is reported to the requested SU in the MCASNs. This notably reduces the average number of sensing bits reported by the selected neighboring SU𝓈 to the fusion center in the MCASNs. In the proposed scheme, each SU after capturing the primary signal sample computes its energy and detects the presence or absence by comparing with the defined threshold value. Then, a one-bit decision 0 for idle and 1 for active with the channel information is sent to the fusion center. However, if no decision is made by the neighboring SU then it simply drops the request without reporting the fusion center. The excessive number of SU𝓈 with no reply may degrade the false alarm probability but the reported local decisions are significantly reduced in the proposed scheme, which saves energy, latency, and processing time consumed in the decision-making process. However, after predefined iterations, if there is no reply for these SU𝓈 in the specific time intervals then they are removed from the channel detection and reporting group list by the fusion center. Consequently, the fusion center combined the neighboring SU𝓈 information and computes the combined likelihood ratio like in a Neyman-Pearson test (T) to make the final decision on the presence or absence of the primary signals which can be numerically indicated as
(13)$$T\left(𝓎\right)=\overset{SU_{𝓃}}{\underset{𝒾=1}{\prod }}\frac{\rho \left({𝓍}_{𝒾}|H_{1}\right)}{\rho \left({𝓍}_{𝒾}|H_{0}\right)}\begin{array}{c} H_{1}\\ ⋛\\ H_{0} \end{array}ℵ\;$$
in which $SU_{𝓃}$ shows that there are 𝓃 number of SU𝓈 and the observations at each SU are indicated by ${𝓍}_{𝒾}$ such that $𝒾=\left\{1,\;2,\;3,\ldots SU_{𝓃}\right\}$ at the fusion center 𝓎 such that $𝓎=\left\{{𝓎}_{1},{𝓎}_{2},{𝓎}_{3},\ldots SU_{𝓃-𝓀}\right\}$ and ℵ is the defined threshold value. This mechanism guarantees the target probability of false alarm while maximizing the probability of detection when making a global decision for the presence or absence of the PU on a channel in the MCASNs. To further enhance the probability of detection, the fusion center adds only those SU𝓈 in a group whose detection probability are higher than others above the defined threshold. The fusion center periodically computes the detection probability of each SU based on its measurements in different rounds and updates its table. Thus, a fewer SU𝓈 with the highest detection probability sense and provide the most accurate results with reduced sensing overhead in UWSNs.

### 3.3. Underwater Acoustic Channel Assignment Algorithm (UWCA)

The SU𝓈 for a channel $C_{i}$ sense the temporal non-existence of the $\;PU_{i}$ and find this channel busy if occupied. The SU𝓈 must leave the detected channel $C_{i}$ and immediately switch to an alternative channel $C_{𝒿}$ by identifying the spectrum holes for transferring packets in UWSNs. The spectrum detecting approaches to find vacant spectrum bands can be categorized into a cooperative (CoS) and non-cooperative (NCoS) in UWSNs. In the CoS approach, the secondary users sense the vacant channels and cooperate closely with the neighboring secondary users to make the decision for the vacant channels in the network. In the NCoS, a secondary user makes the local decision for the vacant channels based on its own spectrum measurement in the UWSNS. The cooperative sensing can be individual (ICoS) or group-based (GCoS) sensing. In the ICoS mechanism, some randomly selected ASNs monitor the data transmission activities of a specific channel independently. By contrast, in the GCoS approach, a set of predefined ASNs is assigned to monitor the activities of a channel in a group. Consequently, the cooperative sensing based on the architecture can be divided into two types, namely centralized cooperative sensing (CCoS) and decentralized cooperative sensing (DCoS) [38]. In the CCoS approach, SU𝓈 for a channel $C_{i}$ sense the temporal nonexistence of the $\;PU_{i}$ and send their local observation data to a specialized fusion center, which combines the received channel results and decides the use of the channel $C_{i}$ for packets transmission. The specialized fusion centers periodically forward the local decision updates to neighboring specialized devices since they are aware of the existing vacant channels at any given time.

Thus, the key aim of the specialized devices is to allocate vacant channels efficiently to SU𝓈 with least information message-sharing in the UWSNs. On the contrary, in the DCoS approach, secondary users sense the vacant channels and cooperate closely with the neighboring secondary users and make the decision without using the centralized device for the vacant channels in the network. The CCoS can considerably intensify the systems aptitude in identifying and evading the PU signals in the network. However, it faces several issues such as high information exchange overheads and particular region cut off issues due to a single expert device failure, and overall network deployment cost due to the high price of these controllers. Thus, it could not be the best choice for the UWSNs. In this respect, the DCoS approach due to its flexibility in deployment for autonomous decision making seems to be the best choice for UWSNs. In DCoS, a set of SU𝓈 without the coordination of the centralized expert devices locally detects the PU signals and maintains the spectrum information to make the decision by itself. In the proposed scheme, the DCoS overcomes the drawbacks of CCoS, but the sensing capabilities of SU𝓈 in DCoS are usually limited due to software-defined hardware limitations such as low computation, signal processing, memory, etc. The DCoS is further divided into restricted spectrum sensing (DRS) and whole spectrum sensing (DWS) [39].

The DRS allows secondary users to sense for the vacant channels in a specific region. Although, secondary users sequentially sense the specific region for available channels might raise the probability of detecting an available channel more quickly. However, the possibility of finding a poor capacity-aware channel with high interference also arises that affects the neighboring ASNs leading to low link qualities in the network. This results in packets loss, significant delay and degrades secondary users’ throughput in the network. On the contrary, the DWS allows secondary users to allow to sense the whole region for the desired channel instead of sensing the specific region. Compared to DRS, the DWS in the proposed scheme increases the chance of finding high data rates channels at low interference, sensing overheads and latency due to allowing various SU𝓈 to sense different channels concurrently in UWSNs. Consequently, there are three types of channel assignment mechanism for multichannel schemes, namely static channel assignment (SCA), dynamic channel assignment (DCA) and semi-dynamic channel assignment (SDA) in UWSNs. In SCA, the acoustic environments such as interferences, data traffic pattern, etc., are generally known [40]. The protocols based on SCA mechanism usually assign channels for long durations relative to the channel switching time or permanently relative to the acoustic interfaces in the network. The static channel assignment is easiest to implement and beneficial to delay-sensitive applications since it avoids additional switching delay during data communication in the network. This type of channel assignment offers several advantages such as low complexity and overhead in the network. However, it is not suitable for the applications whose links are varied and data traffic is unknown due to highly dynamic UWEs.

In UWSNs, the data traffic passing over an ASN differs based on its location along a data path. Generally, the ASNs closer to the sink convey more packets compared to the sensors that are far away. Therefore, data reliability is one of the main requirements for various underwater monitoring and control applications. The static channel assignment mechanisms in underwater suffer from link burstiness due to high interferences and channel variation in the network. This result in packet loss between a sender and receiver for long time intervals, resulting instability and severe communication delays in the scheme. This issue becomes more severe when routing topology varies dynamically. Thus, the SCA limits channel utilization and cannot provide data reliability in the underwater. To this end, the DCA is introduced to mitigate the impact of the interference on the link dynamics due to acoustic channel variations and channel traffic changes in the underwater. The DCA allows the channel assignment schemes to allocate channels with more accurate decisions to secondary users. The DCA provides high throughput in harsh nature dynamic underwater environments. However, it is with the expense of freshness of the data delivery and overhead due to very frequent channel assignment, typically before each transmission. The SDA approach takes the advantages of both static and dynamic channel assignments and provides better trade-off between low overheads and traffic changes with high adaptation to channel variety in the proposed scheme. In this mechanism, the channels are assigned periodically or based on events in the network. Generally, high capacity links with low interferences are allocated statically to the sensors closer to the sink that have the heaviest traffic loads in the network.

The rest of the available channels are allocated for other sensors to pursue a dynamic channel assignment. The semi-dynamic mechanism due to its adaptability to links dynamics, data traffic variations, and interferences while managing the long switching latencies provides a high throughput for delay-sensitive underwater events monitoring applications. To this end, the key objective function of UWCA algorithm can be numerically indicated as

(14)$$\phi_{UWCA\left(2\right)}=\overset{\left|n\right|}{\underset{i=1}{\sum }}\underset{i}{\max\;}\left(C_{𝓊}+\;T_{𝓅}\right)+\underset{i}{\min}\left(I_{𝓃𝓉𝒻}+P_{ℯ𝓇}+C_{𝓉}\right)\;$$

In the proposed scheme, the PU activity information can provide an opportunity to SUs for their reliable data transmission in the network. Therefore, the statistical model of the PU𝓈 behavior must be simple enough that describes precisely how the PU𝓈 are envisioned to operate in the network. The widely considered model that accurately describes the behavior of the PU in an ON–OFF model. In this model, there exist two states, namely ON state and OFF state are used to describe the activity of each channel, indicating the channel is occupied by a PU and the channel is idle, respectively. The SU can use the oFF period of the PU channel to transmit its own data in the network. However, if the OFF period of the PU channel is too small then the spectrum state may not change for the SU𝓈 due to not utilizing the channel in case the primary user is absent. Generally, the exponential model provides a good approximation to measure the duration of the PU states. In this model, the durations of both ON and OFF periods is considered as exponentially distributed and independent with means $1/I_{𝒹\ell ℯ}$ and $1/B_{𝓊𝓈𝓎}$, respectively. The PDF function for both ON and OFF states is numerically computed as:(15)$$F_{B_{𝓊𝓈𝓎}}\left(𝓉\right)=𝒶{ℯ}^{-𝒶\left(𝓉\right)},\;𝓉\geq 0,\;$$
(16)$$F_{I_{𝒹\ell ℯ}}\left(𝓉\right)=𝓫{ℯ}^{-𝓫\left(𝓉\right)},\;𝓉\geq 0,\;$$
in which 𝒶 and 𝓫 represent the behavior of the PU transition rate from ON state to OFF state and from OFF state to ON state, respectively. The channel steady state probability for both the busy and idle can be numerically computed as:(17)$$\rho_{\left(B_{𝓊𝓈𝓎}\right)}=\frac{𝓫}{𝒶+𝓫}\;$$
(18)$$\rho_{\left(I_{𝒹\ell ℯ}\right)}=\frac{𝒶}{𝒶+𝓫}$$
in which $\rho_{\left(B_{𝓊𝓈𝓎}\right)}$ and $\rho_{\left(I_{𝒹\ell ℯ}\right)}$ indicate the busy probability and idle probability, respectively. In the channel-modeling problem of the designed scheme, a suitable channel $C_{i}$ from the available channel list $C_{𝓃}$ is occupied by each secondary user $SU_{\;𝒿}$ opportunistically when a $PU_{𝒾}$ goes silent for $\tau_{\mathrm{off}}$ seconds in order to maximize the total channel utility in UWSNs. In the designed mechanism, the impact of the noise factor $\left(N_{𝑜}\right)$ on the channel selection in a specific time $\tau_{𝒾}$ interval in the underwater environments is computed as:(19)$$N_{𝑜}=\overset{𝓃}{\underset{𝒾=1}{\sum }}{\left(AN_{𝑜}+MN_{𝑜}+TN_{𝑜}+NN_{𝑜}\right)}^{\tau_{𝒾}}$$
in which $TN_{𝑜}$, $AN_{𝑜}$,$\;+NN_{𝑜}$ and $MN_{𝑜}$ are, the thermal noise, the acoustic channel noise factors due to signal internal interferences, nature noise contains factors like seismic, rain, wind and marine animals, and man-made noise like shipping and a motor on a boat in the UWEs. All these factors affect the acoustic signal, which results in reducing the signal quality in UWSNs. In the underwater environment, the signal strength of association between each secondary user $SU_{\;𝒿}$ and, primary or secondary user $\left(PS_{𝒿}\right)$ is highly correlated to the capacity of data transmission of a channel $\left(C_{𝒾}\right)$ in the network. However, occasionally it leads to power spectral density $\left(S_{𝓅𝒹}\right)$ issue in the network. Therefore, the effect of $S_{𝓅𝒹}$ during a channel assignment process is considered as interference for neighboring secondary users in the network. Consequently, a $SU_{\;𝒿}$ during the channel assignment process finds a free channel can only transmit events related information by following the transmission power consist of a feasible range $T_{𝓅\left(𝓃\right)}=\left\{T_{𝓅1},T_{𝓅2},\ldots ,T_{𝓅\left(𝒾\right)}\right\}$ numerically shown as:(20)$$T_{𝓅\left(𝓃\right)}=\overset{𝓃}{\underset{𝒾=1}{\sum }}T_{𝓅\left(C_{𝒾}\right)},\;T_{𝓅\left(C_{𝒾}\right)}\leq \;T_{𝓅\left(𝓉𝒽𝓇ℯ𝓈𝒽𝑜\ell 𝒹\right)};\;$$
subject to:(20a)$$0\leq T_{𝓅\left(min\right)}\leq X\cdot T_{𝓅}{\left(C_{𝒾}\right)}_{l}\leq \;T_{𝓅\left(max\right)}\leq 1\;X\in \left\{0,1\right\}\;,\;l\in L$$

(20b)$$T_{𝓅\left(rec\right)}\left(C_{𝒾}\right)=\frac{T_{𝓅\left(tra\right)}}{T_{𝓅\left(A\right)}N_{f\left(C_{𝒾}\right)}}{\left|G_{\left(C_{𝒾}\right)}\right|}^{2}\;$$

(20c)$$0\leq \underset{l\in L_{i}^{out}}{\sum }T_{𝓅\left(l\right)}\leq T_{𝓅\left(max\right)},\;l\in L\;$$

(20d)$$T_{𝓅\left(E_{CS}\right)}=\left(\underset{l\in L_{i}^{out}}{\sum }T_{𝓅\left(l\right)}+\underset{l\in L_{i}^{In}}{\sum }X_{l}\cdot T_{𝓅\left(rec\right)}\right).\tau_{s},\;l\in L\;$$

(20e)$$B_{try}\left(\tau_{s}+1\right)=\min\left\{B_{max},\max\left[B_{out},B_{i}\left(\tau_{s}\right)+R_{i}\left(\tau_{s}\right)-E_{i}\left(\tau_{s}\right)\right]\right\}\;\tau_{s}\in \;\tau \;$$

(20f)$$\tau_{s}\geq 0\;1<\tau_{s}<\;\tau \;$$

(20g)$$T_{𝓅}\geq 0\;1<B_{out}\leq B_{max}\;$$

(20h)X∈{0,1} 

Constraints in (20a) show that the transmission power for each ASN cannot exceed $T_{𝓅\left(max\right)}$ in the network. Constraints in (20b) indicate the received power $\left(T_{𝓅\left(rec\right)}\right)$ for a particular channel $C_{𝒾}$ with single radio power transmitting antenna $T_{𝓅\left(A\right)}$ in the network. In which $T_{𝓅\left(tra\right)},$
$N_{f\left(C_{𝒾}\right)}$ and $G_{\left(C_{𝒾}\right)}$ are, the transmission power, the number of frequency channels and channel gain in the UWEs. Constraints in (20c) satisfy the transmission power constraint due to several potential outgoing links at acoustic sensor node i in the network. Constraints in (20d) are the energy consumption $\left(E_{CS}\right)$ constraints during a time slot at acoustic sensor node i in the network. Constraints (20e) verify the residual energy of each ASN at the end of time slot $\tau_{s}$ is since the energy level is constrained between battery outage $\left(B_{out}\right)$ and maximum battery power $\left(B_{max}\right)$ in the network. Where the $B_{i}\left(\tau_{s}\right)$, $B_{try}\left(\tau_{s}+1\right)$, $R_{i}\left(\tau_{s}\right)$ and $E_{i}\left(\tau_{s}\right)$ are, the remaining energy of each ASN at the beginning, the remaining energy of each ASN at the end of time slot $\tau_{s}$, the rate of energy replenishment that changes dynamically and initial battery power. Constraints in (20f) to (20h) support constraints from Equations (20a) to (20e). Thus, the high transmission power above the defined low transmission level notably minimizes the effect of $S_{𝓅𝒹}$ for MCASNs in the UWSNs. Note that the $T_{𝓅\left(𝓂𝒶𝓍\right)}\left(C_{𝒾}\right)=T_{𝓅\left(𝓉𝒽𝓇ℯ𝓈𝒽𝑜\ell 𝒹\right)}\left(C_{𝒾}\right)$ is the condition when a secondary user may forward the events information over an allocated channel by using the maximum transmission power when all neighboring nodes go silent in the network. Consequently, by considering the defined transmission power level the SNR must be greater than the defined threshold level $\left(T_{𝓈𝓃𝓇}\right)$ in order to decode received signal correctly can be numerically indicated as:(21)$$SNR_{\left(PS_{𝒿}\right)}=\frac{T_{𝓅\left(tra\right),\left(PU_{𝒾},\;PS_{𝒿}\right)}G_{{\left(C_{𝒾}\right)}^{l}}}{N_{𝑜}+\sum_{k,m\neq i,j}T_{𝓅\left(PU_{k},\;PS_{m}\right)}}\geq T_{𝓈𝓃𝓇}$$

In which $G_{{\left(C_{𝒾}\right)}^{l}}$ is the channel gain of a link ℓ between $PU_{𝒾}$ and $PS_{𝒿}$ in the network. In addition, the capacity of carrying events data $\left(D_{𝒸}\right)$ of a channel $C_{𝒾}$, i.e., $C_{i}\in \underset{\;}{\sum }\left\{C_{1},C_{2},\ldots ,C_{𝓃}\;\right\}$ is one of the main criteria to satisfy the constraints of the events monitoring in UWSNs. Therefore, the constraints on data transmission with different rates $\left(D_{𝓇}\right)$ depends on the transmission power and frequency channel selection of a $SU_{\;𝒿}$ in the network. This can be computed numerically by using Equations (22) and (23) as:(22)$$D_{𝓇}{}_{\left(C_{𝒾}\right)}=B_{\ell }log_{2}\left(1+\frac{G_{{\left(C_{𝒾}\right)}^{l}}\;T_{𝓅\left(C_{𝒾}\right)}}{N_{𝑜}}\right)\;l\in SU_{\;𝒿},\;PS_{𝒿};\;B_{\ell }\in N_{f\left(C_{𝒾}\right)}$$

Accordingly, when $T_{𝓅\left(max\right)}$ is used, the corresponding maximum data rate is:(23)$$D_{𝓇}{}_{\left(max\right)}\left(C_{𝒾}\right)=B_{\ell }log_{2}\left(1+\frac{G_{{\left(C_{𝒾}\right)}^{l}}\;T_{𝓅\left(max\right),\left(C_{𝒾}\right)}}{N_{𝑜}}\right)\;l\in SU_{\;𝒿},\;PS_{𝒿};B_{\ell }\in N_{f\left(C_{𝒾}\right)}$$
in which the $B_{\ell }$ is the bandwidth of a link ℓ between $SU_{\;𝒿}$ and $PS_{𝒿}$ in the network. Thus, by considering the above data transmission rates with high transmission power, the average throughput $\left(T_{𝒽𝓅}\right)$ of the 𝒾th channel $T_{𝒽𝓅}\left(C_{𝒾}\right)$ during sensing time $\tau_{𝒾}$ from the total sensing time slot τ is computed as:(24)$$T_{𝒽𝓅}\left(C_{𝒾}\right)=\rho_{\left(H_{0},C_{𝒾}\right)\;}\left(1-\rho_{\left(f𝒶,\;C_{i}\right)}\left(\tau_{𝒾},\sigma \right)\right)+\rho_{\left(H_{1},C_{𝒾}\right)\;}\left(1-\rho_{\left(𝒹,\;C_{i}\right)}\left(\tau_{𝒾},\sigma \right)\right)\;$$

However, this high transmission power significantly improves the data transmission rate to a faraway distance in the network. However, it brings severe interference to neighboring sensors in the UWSNS. This interference $\left(I_{𝓃𝓉𝒻}\right)$ constraint due to high transmission power in the underwater can be computed by using the follow Equation (25) as:(25)$$I_{𝓃𝓉𝒻}\left(𝒸𝑜𝓃𝓈\right)=\overset{𝓃}{\underset{𝒾=1}{\sum }}\overset{𝓂}{\underset{𝒿=1}{\sum }}{\left(\frac{T_{𝓅\left(SU_{\;𝒿}\right)}\;\cdot }{\sum T_{𝓅\left(\left(PS_{𝒿}\right)\right)}\cdot G_{\left(SU_{\;𝒿},\;PS_{𝒿}\right)}}\right)}^{C_{𝒾}}$$
and the interference range of a secondary user $R_{I_{ntf}}\left(SU_{\;𝒿}\right)$ numerically written as:(26)$$R_{I_{𝓃𝓉𝒻}}\left(SU_{\;𝒿}\right)=\overset{𝓃}{\underset{𝒾=1}{\sum }}\overset{𝓂}{\underset{𝒿=1}{\sum }}\frac{PU_{𝒾,\;P_{\ell 𝑜𝓈𝓈}\left({𝒹}_{𝓂𝒶𝓍}\right)}}{SU_{\;𝒿,P_{\ell 𝑜𝓈𝓈}\left(𝒹\right)}\;+B_{I_{𝓃𝓉𝒻}\;𝓅\left(𝓂𝒶𝓍\right)}\left(\;C_{𝒾}\right)}\;$$

The SU by using signal level measurements must estimate the interference level, it creates at the PU receivers during continuous transmission in UWSNs. The SU and PU could transmit data simultaneously only if they are far from each other and have low interference effect computed by the signal-to-interference ratio $\left(S_{𝒾𝓇}\right)$ in the UWSNS. The interference range based on the minimum distance (D) information between the SU transmitter and PU receiver away from each other can be computed as:(27)$$S_{𝒾𝓇}\left(D\;\right)=\frac{P_{PU_{𝒾}}^{{}^{\prime }}\cdot G_{{\left(C_{𝒾}\right)}^{l}}\left({𝒹}_{1}^{{}^{\prime }}\;\right)}{P_{SU_{\;𝒿}}^{{}^{\prime }}G_{{\left(C_{𝒾}\right)}^{l}}\left({𝒹}_{2}^{{}^{\prime }}\;\right)+P_{𝓫}^{{}^{\prime }}}\;$$
in which ${𝒹}^{{}^{\prime }}\;$is the distance between transmitter and receiver acoustic nodes, $P_{𝓫}^{{}^{\prime }}$ at the primary receiver $P_{PU_{𝒾}}^{{}^{\prime }}$ and $P_{SU_{\;𝒿}}^{{}^{\prime }}$ is the power of background interference, ${𝒹}_{1}^{{}^{\prime }}$ is the distance information of a PU located away to the PU transmitter and ${𝒹}_{2}^{{}^{\prime }}$ is the distance information of a secondary user located away to the primary user receiver in the network. The SU must be capable of detecting the PU transmitter signals within the range of ${𝒹}_{1}^{{}^{\prime }}\;$+ ${𝒹}_{2}^{{}^{\prime }}$, which translates to a sensitivity constraint for the secondary user detector to prevent harsh interference from the primary user receiver in the network. Thus, to protect the active PUs for their reliable events data transmissions to neighboring SUs over a channel $C_{𝒾}$, the interference constraint from all opportunistic transmissions must not exceed a predefined tolerable threshold in the network. It is assumed that each $SU_{\;𝒿}$ selects an appropriate data capacity
channel from the existing channel list i.e.,$\;SU_{\;𝒿}\left(C_{𝒾}\right)\cap PS_{𝒿}\left(C_{𝒾}\right)=\;\emptyset$, i.e., $f\left(SU_{\;𝒿}\right)+f\left(PS_{𝒿}\right)\leq 1$ in which $f\left(PS_{𝒿}\right)=1/\underset{\;}{\sum }\left\{C_{𝓃}\;\right\}$ and $f\left(PS_{𝒿}\right)=1/\underset{\;}{\sum }\left\{C_{𝓃-1}\;\right\}$ for $\left\{f\in C_{𝓃}\right\}$ is the normalized frequency bands allocated to a set of distinct acoustic sensors in the network. Thus, for each secondary user, the constraint is to limit all neighboring acoustic sensor nodes of a secondary user for using different spectrum channels for avoiding the interference effect in the UWSNs, which are satisfied by the constraints in Equation (28) can be numerically written as:(28)$$I_{𝓃𝓉𝒻}=\overset{𝓃}{\underset{𝒾=1}{\sum }}\overset{𝓂}{\underset{𝒿=1}{\sum }}G_{{\left(PU_{𝒾},\;PS_{𝒿}\right)}^{C_{𝒾}}}\leq \;T_{I_{𝓃𝓉𝒻\left(PS_{𝒿}\right)}}\leq 1\;,\;\forall \;PU_{𝒾}\in C_{𝓃}\;$$
subject to:(28a)$$I_{𝓃𝓉}\left(PS_{𝒿}\left(C_{𝒾}\right)\right)\in R_{𝒾}\leq I_{𝓃𝓉𝒻}\left(\tau_{s}\right)<1\;$$

(28b)$$E_{𝓉𝓃}\left(PS_{𝒿}\left(C_{𝒿}\right)\right)\in R_{𝒿}\leq I_{𝓃𝓉𝒻}\left(\tau_{s}\right)<1\;$$

(28c)$$I_{𝓃𝓉}\left(PS_{𝒿}\left(C_{𝒾}\right)\right)\in R_{𝒾}|R_{𝒿}\leq I_{𝓃𝓉𝒻}\left(\tau_{s}\right)\leq 1\;$$

(28d)$$E_{𝓉𝓃}\left(PS_{𝒿}\left(C_{𝒾}\right)\right)\in R_{𝒿}|R_{𝒾}\leq I_{𝓃𝓉𝒻}\left(\tau_{s}\right)\leq 1\;$$

(28e)$$I_{𝓃𝓉}\left(PS_{𝒿}\left(C_{𝒾}\right)\right)\in R_{𝒿}\cup R_{𝒿}\leq I_{𝓃𝓉𝒻}\left(\tau_{s}\right)\leq 1\;$$

(28f)$$PU_{𝒾}\cap PS_{𝒿}\left(C_{𝒾}\right)\in R_{𝒾}\cup R_{𝒿}\neq C_{𝒾}\;$$

(28g)$$PU_{𝒾}\cap PS_{𝒿}\left(C_{𝒾}\right)\in R_{𝒾}\cup R_{𝒿}=C_{𝓀},\;C_{𝓀}\subseteq \;C_{𝓂𝒶𝓍}\;$$

(28h)$$\forall \ell_{𝒿}\left(C_{𝒾}\right)\leq C_{𝓀}\;\leq 1;\;\forall \;PS_{𝒿}\in \left(R_{𝒾},R_{𝒿}\right),\;\forall \ell_{𝒿}\in \;L\;$$

(28i)$$\ell_{𝒿}\left(C_{𝒾}\right)>0;\;R_{𝒾}\cup R_{𝒿}\geq 0;\;I_{𝓃𝓉𝒻}\geq 0$$

(28j)$$\tau_{s}\geq 0\;;\;1<\tau_{s}<\;\tau \;$$

Constraints in (28a) and (28b) ensure that the internal $\left(\;I_{𝓃𝓉}\right)$ and external $\left(E_{𝓉𝓃}\right)$ interferences of the primary or secondary users belong to a different regions $R_{𝒾}$ and $R_{𝒿}$ using different frequency channels $C_{𝒾}$ and $C_{𝒿}$ at time $\tau_{s}$ is less than the defined threshold level. Constraints in (28c) and (28d) guarantees that the $I_{𝓃𝓉}$ and $E_{𝓉𝓃}$ of the primary or secondary users belong to the different regions $R_{𝒾}$ or $R_{𝒿}$ using the same frequency channels $C_{𝒾}$ at time $\tau_{s}$ must be less than the defined threshold level. Constraints in (28e) state that the $I_{𝓃𝓉𝒻}$ for the primary or secondary users belong to a region $R_{𝒾}$ or $R_{𝒿}$ using the same frequency channel at time $\tau_{s}$ must not be greater than the defined threshold value. Constraints (28f) is the supporting constraints for (28e) guarantees that at the same time $\tau_{s}$ primary and secondary users in a region $R_{𝒾}$ or $R_{𝒿}$ cannot use the same channel $C_{𝒾}$ to avoid harmful interference while the constraints in (28g) confirm that different channels are assigned to each primary and secondary users in the network. Constraints from (28h) to (28j) are the supporting constraints for Equation (28a) to (28g). Further, to maximize the channel utilization with minimum interference effect during channel selection in a region $R_{𝒾}$ in UWSNs. Considering the interference issues, the maximum utility for each user’s channel selected for the information transmission is computed numerically in Equation (29) as

(29)$$U_{𝓉}=\underset{\;U_{𝓉}\left(SU_{\;𝒿}\right)\cdot \left(C_{𝒾}\right)}{\max}\overset{n}{\underset{i=1}{\sum }}\phi_{SU_{\;𝒿}}\cdot \mu \left(\varphi_{C_{𝒾}}^{SU_{\;𝒿}}\left(U_{𝓉\left(SU_{\;𝒿}\right)}\right)\right)\;$$

This maximum utility function, i.e., $\phi_{SU_{\;𝒿}}\left(\varphi_{C_{𝒾}}^{SU_{\;𝒿}}\right)=\phi_{SU_{\;𝒿}}\log\left(1+\varphi_{C_{𝒾}}^{SU_{\;𝒿}}\;\right)$ is assumed to be proportional to the Shannon capacity biased by priority parameter $\phi_{SU_{\;𝒿}}$ of the secondary user $SU_{\;𝒿}$ by using channel $C_{𝒾}$ in the network. As stated above, to achieve a longer idle probability for SUs, the desired channels are assigned by considering the previous channels history values saved in the channel table. The proposed scheme to achieve high throughput and data rates for longer time, arranges the channel information in the table in a systematic manner. In fact, it sorts channels in a way that a channel with high achievable throughput and data rate available for a longer time has high priority in the channel table. The priority of the channels periodically monitors and changes in each round of the data transmission in the network. Consequently, the primary users channel history $P_{C\left(𝒽\right)}$ based on the sensing results is maintained by each secondary user node $SU_{\;𝒿}$ with a predefined priority $\left({𝓅}_{𝒾}\right)$ in the underwater, which can be estimated as:(30)$$P_{C\left(𝒽\right)}\left(SU_{\;𝒿}\right)=\overset{𝓃}{\underset{𝒾=1}{\sum }}\overset{\ell }{\underset{𝓀=1}{\sum }}\;{𝓅}_{C_{𝒾}}{\left(PU_{𝒾}\right)}^{𝒽\left(𝓀\right)}\;$$

On the other hand, during the time $\tau_{𝒾}$ the channels stored with decreasing priority in a $PS_{𝒿}$ sensor’s channel table is numerically estimated by using the following Equation (31):(31)$$PS_{𝒿}^{𝒽\left(𝓀\right)}=\overset{𝓃}{\underset{𝒾=1}{\sum }}\overset{\ell }{\underset{𝓀=1}{\sum }}\;{\left(PU_{𝒾}\right)}^{𝒽\left(𝓀\right)}\;$$

The volume of channel history data stored $\left(D_{𝓈}\right)$ in the channel table of a secondary user at a given time $\tau_{𝒾}$ is numerically indicated as:(32)$$D_{𝓈}\left(SU_{\;𝒿}\right)=\overset{\tau_{0}}{\underset{i=0}{\sum }}D_{𝓈}{\left(SU_{\;𝒿}\right)}^{C_{𝒾}}\left(\tau \right)d\tau$$

In the proposed scheme, a primary or the secondary user sensor may share information to its single hop neighboring sensors and sense the entire region only in a case where the desired capacity channel is currently occupied or does not have desired capacity channel information stored in the CIT. Consequently, at each iteration of the negotiation procedure, the sensor’s CIT is periodically updated in the MCASNs. In the proposed scheme, we employ First in First Out (FILO) policy to avoid the management complexity of the events data in a channel information table (CIT). Thus, a channel that is not used in a predefined time is substituted with the new one in the channel table of a primary or secondary user as shown in Figure 4. In addition, a PU channel with poor data transmission performance is blocked by the SU for a defined time interval to provide the opportunity of data transmission for other SU𝓈 as revealed in Figure 4. It dipicts that the $PU_{9}$ channel is blocked for a predefined time and assigned a lower priority in the CIT since it performs poorly in transmission performance in the UWSNs. Consequently, the probability of ideal channel availability for a secondary user based on the previous channel history between the time $\tau_{𝒾}$ and $\tau_{𝒿}$ for $\tau_{𝓀}$ time can be computed as:(33)$$\rho_{available}^{C_{𝒾}}\left(SU_{\;𝒿}\right)=1-\rho_{\tau_{𝑜𝓃\to 𝑜ff}}^{C_{𝒾}}\left(PU_{𝒾}\right)$$

Thus, the designed scheme saves a notable amount of spectrum re-sensing energy by avoiding the data packet collision due to assigning a longer idle channel between the PUs and SUs in the UWSNs.

### 3.4. Packets Forwarding Scheme (UWPF)

Recently, the designed packets forwarding schemes during packets forwarding prefers to select the shortest forwarder node along a routing path (see Section 2 for details). However, the shorter distance routing mechanism cannot perform well since it increases the number of switching channels in MCASNs. This increases the residual energy consumption and the probability of packet loss at each hop due to high packets collision caused by message overheads in the channel assignment process in the network [24]. In addition, shortest path routing increases the probability of congestion occurrences when the offered data traffic load rises above the node’s current buffer capacity in UWSNs. Therefore, the buffer occupancy frequently needs to be monitored in order to detect incipient congestion in the network. In addition, the number of hops the packets travel also must be reduced to minimize routing delay and improve the data delivery ratio in the network [26]. To this end, the designed packets forwarding scheme consists of four main phases, namely (i) neighboring discovery, (ii) route construction with optimal forwarders, (iii) reliable packets forwarding process and (iv) routes reconstruction procedure in UWSNs. In the proposed mechanism, if an ASN has packets to forward via multi-hop communication to the sea surface sink, it appoints the next-hop forwarder ASNs based on its local knowledge which depends on various parameters of the neighboring relay ASN in the network. The key aims during the packet forwarding procedure, including maximizing the packet delivery ratio $\left(P_{𝒹𝓇}\right)$, minimize the congestion $\left(C_{𝑜𝓃ℊ}\right)$, latency $\left(D_{ℯ\ell }\right)$ and load balancing $\left(L_{𝓫}\right)$ in UWSNs. In this respect, the key objective function of the packets forwarding $\left(\phi_{UWPF\left(3\right)}\right)$ algorithm is numerically indicated as:(34)$$\phi_{UWPF\left(3\right)}=\overset{\left|n\right|}{\underset{i=1}{\sum }}\underset{i}{\max}P_{𝒹𝓇}+\underset{i}{\max}L_{𝓫}+\underset{i}{\min}C_{𝑜𝓃ℊ}+\;\underset{i}{\min}D_{ℯ\ell }\;$$

#### 3.4.1. Efficient Neighboring Discovery Process

In this phase, initial neighboring information tables are constructed, which are used in the routing phase where different data paths are selected in the final routing tables based on the priority of data-forwarding nodes. The neighboring discovery process finds and maintains the updated neighboring ASNs information for each ASN in UWSNs. Initially, a remote user initiates the neighbors finding process by forwarding a network initialization (𝒾𝓃𝒾_𝓂𝓈ℊ) message to the sea surface sink. Upon receiving the 𝒾𝓃𝒾_𝓂𝓈ℊ message, the sea surface sink broadcasts neighboring discovery (𝓃ℯℊ_𝓂𝓈ℊ) message to randomly deploy acoustic nodes in its communication range with the following values: the sink identity, level number and location information with the angle of packet departure in UWSNs. An acoustic node closest to the sink receives the 𝓃ℯℊ_𝓂𝓈ℊ message successfully, sets the layer number to maximum 𝓃, which infers that it has a direct contact with the sea surface sink. Then, each ASN creates a new record entry for neighboring sink and computes the distance to the sea surface sink of itself and stores this information in the neighboring information table. The ASNs directly received the 𝓃ℯℊ_𝓂𝓈ℊ messages from the sea surface sink are marked as level 𝓃. This level is periodically decreasing in the downward direction for the nodes located in the lower layers such as 𝓃−1, 𝓃−2, etc., and reaches to 0 as shown in Figure 5a. After updating the sink record, these ASNs rebroadcasts the 𝓃ℯℊ_𝓂𝓈ℊ message to their neighbors within their transmission radius with the following values: the sender identity, current channel, remaining energy, level number, position, angle of departure and distance to the sink. Upon receiving the 𝓃ℯℊ_𝓂𝓈ℊ message correctly, each ASN creates a record entry for new neighboring ASNs then computes the distance to the sender of itself, angle of packets arrival, level number, and stores the entire received information in the neighboring information table. This procedure is repeated until each ASN has its neighbors information stored in the routing table. During each message exchange process, the receiver acoustic node sends an acknowledgment message to the sender using the CSMA mechanism to ensure the guaranteed delivery. Thus, the communication links between acoustic nodes are bidirectional tested in UWSNs. At this stage, each acoustic node is aware of its neighbors deployed for events monitoring in UWSNs.

#### 3.4.2. Routes Construction with Optimal Forwarders

The route construction process starts from the source node have packets to convey to the sea surface sink. To do so, the source node by considering the local records sends the route discovery (𝓇𝒹𝒾𝓈_𝓂𝓈ℊ) message to its neighbors with the following values: the sender identity, current channel, remaining energy, level number, position with the angle of message departure and distance to the sink. Upon receiving the 𝓇𝒹𝒾𝓈_𝓂𝓈ℊ message successfully, each ASN computes and updates the distance to the sender of itself, angle of packet arrival, level number in the routing table. The receiver node compares its own distance and the sender distance to the sink. Then, it sends its own information of sender identity, current channel, remaining energy, level number, position with the angle of message received with value 1 only if its distance to the sink is lower than the sender, otherwise it is 0. The receiver node also computes the priority of the sender node and updates in the routing table with decreasing priority. In fact, this priority is important for sender since it will be used during reverse routes finding towards the data source. Upon receiving the reply (𝓇ℯ𝓅_𝓂𝓈ℊ )message correctly, the source node computes the angle of packets arrivals and updates the sender’s received information with a priority in the routing table. Usually, a candidate with high residual energy, lower distance to the source node and sink is given high priority in the routing table. Consequently, the priority of each neighboring acoustic node in both upwards and downwards direction is updated in the routing table. In addition, the forwarding candidates with the same level 𝓃 and being equal to the sink have the intermediate priority in the routing table. The forwarding candidates with the lowest level (𝓃−𝓀) such that 𝓀∈{1,2,…,𝓃} and long distance has the lowest priority in the routing table since they are far away to the sea surface sink.

The intermediate and lowest priority forwarding candidates are selected as forwarding candidates only if a suitable forwarding node is not found in the transmission range of the source node in the network. Both the intermediate and lowest priority candidates are usually called helpers or guide nodes in the network. Thus, a node with the highest priority in the routing table is selected as a potential forwarder to relay information of the source node towards the sink. The higher the priority, the shorter is the time for data to reach the sea surface sink. In the packet forwarding process (see Section 3.4 for details) it is also possible that a huge volume of events data is moved over a particular forwarder result in packets overflow problem due to the limited buffer size. Therefore, after a predefined iteration, a new parameter called buffer overflow time is included in the priority list of each potential forwarder to avoid congestion in UWSNs. To prevent congestion occurrence, each forwarder node monitors its buffer occupancy level periodically since it is gradually filling up. The congestion is detected once the buffer occupancy exceeds a predefined threshold value in the network. Then, the congestion avoidance procedure begins, which is based on diverting the incoming data traffic to the other available forwarding candidates towards the sea surface sink. This reduces packets loss rate and thus help to maximize the overall packet delivery ratio in the network. During each message exchange process, the receiver acoustic node sends an acknowledgment message to the sender to ensure the guaranteed delivery. Subsequently, the route discovery process at each hop follows the same previous procedure until the forwarding candidate closer to the sea surface sink is found to convey packets. In the worst case, if a suitable packets forwarder is not found in the sender’s communication range in upward direction then it sends 𝓇𝒹𝒾𝓈_𝓂𝓈ℊ to the same or lower levels neighboring candidates. Upon receiving the 𝓇𝒹𝒾𝓈_𝓂𝓈ℊ, each receiving acoustic node located on the same or lower levels marks itself as a helper node. The more potential helpers in the same or low level increases the opportunity to find a relay node with the highest priority on the network. The receiving node repeats the same aforesaid procedure to find the most appropriate forwarder towards the sea surface sink.

Finally, the route discovery message is delivered to the sea surface sink, which sends an 𝒶𝒸𝓀_𝓂𝓈ℊ message to the downstream senders, including the identity of each forwarding relay node which is delivered to the source node. An acoustic relay node that receives an 𝒶𝒸𝓀_𝓂𝓈ℊ message from the sea surface sink first it checks and decides either it was the selected next-hop forward along a distinct routing path in the upstream direction. If this is the case, the relay node confirms that it is on the reverse path to the source node and marks itself as a reverse forwarder candidate in the network. Then, the forwarder candidate updates its records and after setting the priority in the routing table forwards the received message to its downstream link candidates located on the lower levels. In this way, the reverse route construction information is propagated by each relay node until it reaches the data source node in the network. Finally, this process finds reverse paths with guaranteed upward routing paths from the source to the destination. In this entire process, if a receiver candidate receives the same request messages multiple times from the same sender node, the destination node shall only reply to the first received request message and neglect others. In addition, each forwarder maintains two tables of best forwarding candidates upwards with high priority and downwards with low priority, respectively. This entire mechanism ensures that the packets will travel over a set of optimal forwarders with the appropriate distances before getting to the sea surface sink. The packets forwarding along a restricted narrow pipe like routing path by considering the acoustic nodes minimum angle information significantly minimize the average routing path length as shown in Figure 5a,b. Thus, a set of high quality routes from source to the destination are constructed in both upward and downward directions for reliable data transmission in the network. Thus, the chance of data path loops and congestion occurrences are reduced notably in the network.

#### 3.4.3. Reliable Packets Forwarding Process

At this stage, each source node has the multiple constructed data paths information towards the sea surface sink. The source node selects the best path among the available and sends a ready data (𝓇ℯ_𝒹𝒶𝓉𝒶) to neighbors and the potential forwarder, so that it can tune its channel to the requested one for receiving data packets. The 𝓇ℯ_𝒹𝒶𝓉𝒶 message includes the sender identity and channel information. This mechanism avoids excessive acknowledgment message sharing to the individuals and thus minimize energy consumption, but also provides information to neighbors about the usage of an occupied channel in the network. The packets transmission begins as soon as the acknowledgment message is received from the receiver in the network. This procedure repeats at each hop until the data packets are successfully delivered to the sea surface sink. After receiving the packets, the sink sends a confirmation message to the senders in the reverse routing path to notify the source node that its data has been received for further processing and sent to a remote user in the network.

In addition, the similar procedure is repeated in the reverse packets forwarding from the sink towards the source node in UWSNs. Thus, the entire data packets are greedily forwarded to the neighbors’ relay candidates closest to the destination in a multi-hop manner. Finally, the entire gathered data is upload from the source to the destination over the best upstream and downstream multi-hop links over a set of best forwarding nodes, which numerically can be expressed as:(34a)$$D_{𝒾\to 𝒿}=D\left(𝒾,S_{𝒾𝓃𝓀}\right)-D\left({\left(𝒿\right)}_{𝓇ℯ\ell 𝒶𝓎},S_{𝒾𝓃𝓀}\right)\leq 1\;\forall \;𝒾,{𝒿}_{𝓇ℯ\ell 𝒶𝓎}\in \;S_{𝒾𝓃𝓀}$$

(34b)$$D_{𝒿\to S_{𝒾𝓃𝓀}}=\;D\left(F_{𝓇ℯ\ell 𝒶𝓎\left\{\left(\left(𝒿,𝓀\right)\subseteq 𝓃\right)\right\}}\right)>D_{𝓂𝒾𝓃}\;\&\&\leq D_{𝓂𝒶𝓍}\;\forall {𝓀}_{𝓇ℯ\ell 𝒶𝓎}\in \;S_{𝒾𝓃𝓀}$$

(34c)$$D_{𝒿\to S_{𝒾𝓃𝓀}}=\;D\cdot F_{𝓇ℯ\ell 𝒶𝓎\left(𝓃\right)}\;\in \;U{𝓅}_{𝓈𝓉𝓇ℯ𝒶𝓂}\;\forall F_{𝓇ℯ\ell 𝒶𝓎\left(𝓃\right)}\in \;S_{𝒾𝓃𝓀}$$

(34d)$$D_{S_{𝒾𝓃𝓀}\to 𝒾}=\;D\cdot RF_{𝓇ℯ\ell 𝒶𝓎\left(𝓃\right)}\;\in \;D𝑜𝓌{𝓃}_{𝓈𝓉𝓇ℯ𝒶𝓂}\;\left(1q\right)\;\forall \;BF_{𝓇ℯ\ell 𝒶𝓎\left(𝓃\right)}\in 𝒾\;$$

(34e)$$D_{𝓈𝓊𝓂}=\;\underset{𝒿\in F_{𝓇ℯ\ell 𝒶𝓎}>1}{\sum }D_{𝒿\to S_{𝒾𝓃𝓀}}\left(F_{𝓇ℯ\ell 𝒶𝓎\left(𝒿\right)}\right)\;\leq F_{𝓇ℯ\ell 𝒶𝓎\left(𝓂𝒶𝓍\right)}\;\forall {𝒿}_{𝓇ℯ\ell 𝒶𝓎}\in \;S_{𝒾𝓃𝓀}$$

(34f)$$\rho_{b\left(𝓅𝒹\left(𝒾\right)\right)}\left(RP_{𝒾}\right)=\beta_{1}\cdot Q{𝓋}_{𝒾}-\beta_{2}\cdot \left(1-\overset{𝓃}{\underset{𝒿=1}{\sum }}\frac{𝒿\left(Q_{𝒿}\right)}{Q_{\ell }}\right)+\rho_{b}\left(𝓅𝒹{\left(𝒾\right)}_{𝒾𝓃𝒾}\right)\;\forall {𝒿}_{𝓇ℯ\ell 𝒶𝓎}\in \;S_{𝒾𝓃𝓀}$$

(34g)$$\rho_{b}\left(𝓅𝒹{\left(𝒾\right)}_{𝒾𝓃𝓀}\right)=\;\left\{0,\;\mathrm{if}:𝒾Q_{𝒾}<1;\;N{\left(𝒿\right)}^{2}\cdot \left(Q_{𝒿}/Q_{\ell }\right)-1,\;\mathrm{if}:𝒾Q_{𝒾}\geq 1\right\}$$

(34h)$$Q{𝓋}_{𝒾}=\frac{Q_{𝒾}^{𝓃ℯ𝓌}-Q_{𝒾}^{𝑜\ell 𝒹}}{Q_{\ell }/Nℯ\left(𝒿\right)}\;0<\forall Q_{𝒾}\leq Q_{\ell }=1$$

(34i)$$0\leq \rho_{b\left(𝓅𝒹\left(𝒾\right)\right)}<1$$

(34j)$$D_{𝒿\to S_{𝒾𝓃𝓀}}=D\left(𝒿,S_{𝒾𝓃𝓀}\right)/A_{𝓋ℊ}\left(F_{𝓇ℯ\ell 𝒶𝓎\left(𝒿\right)}\right)\;\mathrm{where}\;\forall \;{𝒿}_{𝓇ℯ\ell 𝒶𝓎}>1\;\&\&\;{𝒿}_{𝓇ℯ\ell 𝒶𝓎}\in \;S_{𝒾𝓃𝓀}$$

(34k)$${𝓉}_{𝓀}=\;\overset{𝓃}{\underset{𝒾=1}{\sum }}{𝓉}_{𝓫𝒶𝒸𝓀𝑜𝒻𝒻}+F_{𝓇ℯ\ell 𝒶𝓎}+{𝓉}_{𝒶𝒸𝓀}\leq 1\;\forall L_{ASN\left(𝒾,𝒿\right)}\in \;S_{𝒾𝓃𝓀}$$

Constraints in (34a) state that in single-hop packet progress, the Euclidian distance (D) of next-hop forwarder 𝒿 receiving data packets from an acoustic node 𝒾 must be less than the sender to the sink. The constraints in (34b) states that the forwarding node distance must be greater than the minimum $\left(D_{𝓂𝒾𝓃}\right)$ and less than maximum Euclidian distance $\left(D_{𝓂𝒶𝓍}\right)$ in the network. Constraints in (34c) show that the maximum and minimum distance information of the forwarders during data transmission is bounded to the upward direction $U{𝓅}_{𝓈𝓉𝓇ℯ𝒶𝓂}$ towards the sink. Similarly, the constraints in (34d) verify that the message forwarding from the sink towards the source node 𝒾 in the reverse forwarding nodes $RF_{𝓇ℯ\ell 𝒶𝓎\left(𝓃\right)}$ is bounded to downward direction $\left(D𝑜𝓌{𝓃}_{𝓈𝓉𝓇ℯ𝒶𝓂}\right)$. Constraints in (34e) guarantees that the number of forwarding nodes cannot be higher than the defined maximum forwarding nodes $F_{𝓇ℯ\ell 𝒶𝓎\left(𝓂𝒶𝓍\right)}$ along a routing path towards the sink. Equation (34f) indicates the probability of a packet drop $\rho_{b}\left(𝓅𝒹\left(𝒾\right)\right)$ in the 𝒾th queue of a forwarding candidate along the selected routing path $RP_{𝒾}$ in the network. In Equation (34f), the $\rho_{b}\left(𝓅𝒾{\left(𝒾\right)}_{𝒾𝓃𝒾}\right)$ and $Q{𝓋}_{𝒾}$ are, the initial packet loss probability and the level of variation in the length of the 𝒾th virtual queue can be determined using Equation (34g) and Equation (34h), respectively. The expression $\overset{𝓃}{\underset{𝒿=1}{\sum }}𝒿\left(Q_{𝒿}\right)/Q_{\ell }$ indicates the used queue space of a forwarding candidate in the network. The $\beta_{1}$, $\beta_{2}$, $Q_{𝒿}$, $Q_{\ell }$ are, the constant depends on the variation in the length of the 𝒾th queue, the queue capacity constant defined by the user, the number of packets in the 𝒿he queue and maximum queue length ℓ of a forwarding node in the network. The $N{\left(𝒿\right)}^{2}$ and $𝒾Q_{𝒾}$ in Equation (34g) are, the number of node’s neighbors and queue length of 𝒾he forwarding node in the network. In Equation (4a), the $Q_{𝒾}^{𝓃ℯ𝓌}$ and $Q_{𝒾}^{𝑜\ell 𝒹}$ is the 𝒾he queue length in the 𝒾he flow in the present and previous calculation in the network. The Equation (34f) is basically used as a congestion indicator at each hop along a routing path from the source towards the destination in the network. The congestion indicator values vary between 0 and 1 for each node in the network. The constraints in (34h) and (34i) indicate that the congestion indicator values always must be less than 1, which means that the queue length is suitably managed when the senders sending the packets. On the contrary, if the congestion indicator value is equal to 1 it means that the queue length of a forwarder is not suitably managed and continue receiving packets from the neighbors will lead to packet loss due to buffer overflow. Thus, before reaching the defined threshold, the priority value of the congestion is decreasing at each forwarding node, which prevents the packets from being dropped. Constraints in (34j) indicate the expected delay from relay node 𝒿 towards the $S_{𝒾𝓃𝓀}$ is computed by the average remaining hop counts $A_{𝓋ℊ}{\left(\right)}_{𝓇ℯ\ell 𝒶𝓎}$ along a routing path in the network. All above constraints guaranteed that the data path loop does exist along a routing path during the packets transmission in both upward and downward directions in the network. Constraints in (34k) show that the back off time to acquire the channel $\left({𝓉}_{𝓫𝒶𝒸𝓀𝑜𝒻𝒻}\right)$ for finding an appropriate forwarding relay node $\left(F_{𝓇ℯ\ell 𝒶𝓎}\right)$ and acknowledgement time $\left({𝓉}_{𝒶𝒸𝓀}\right)$ for sending a data packet must be less than the defined threshold.

#### 3.4.4. Routes Reconstruction Procedure

Generally, the forwarders switch to sleep mode if they do not have more data to transmit to the sink. On the other hand, continue data transmission for a long time can drain relay nodes energy, particularly for those who are nearer to the sink. Moreover, in most cases these relay nodes are facing congestion issues because they are forwarding a huge amount of data coming from the rest of the network. In addition, the possibility of transmission failures cannot be ignored since the wireless links are unreliable in the harsh UWEs. In the first and last cases, if the receiver relay node did not reply to the sender in a predefined amount of time and then it sends the request again to the sender and wait for a predefined amount of time longer than the previous waiting time. The receiver node is declared as an inactive forwarder if the sender did not receive any reply from the destination node. In the second case, the congested node first set up the time and then sends a buffer overflow (B𝑜𝒻_𝓂𝓈ℊ) to its neighbors nodes. The nodes correctly received the B𝑜𝒻_𝓂𝓈ℊ messages set the sender node buffer overflow priority to zero and update the entries in their routing table. This means that the forwarding node currently is not available for packets transmissions in the network. Then, the source node based on its local information launches another route-discovery process in the affected region in the network. In that case, the sender sends a route discovery message to the second highest priority node in the routing table in the upward direction and waits for the predefined time.

The packets transmission begins as soon as the acknowledgment message is received from the receiver in the network. The buffer overflow priority of the selected destination relay node is set to one in the routing table since it is ready to receive packets. Finally, the data with the new relay nodes information is delivered to the sea surface sink, which sends an 𝒶𝒸𝓀_𝓂𝓈ℊ message to the downstream senders to inform them about this new route, including the identity of each forwarding and new selected relay node, and delivers it to the source node. Then each new acoustic relay node which receives an 𝒶𝒸𝓀_𝓂𝓈ℊ message from the sea surface sink first checks and decides either it was the selected next-hop forwarder along a distinct routing path in the upstream direction. If this is the case, the relay node confirms that it is on the newly constructed reverse path to the source node and marks itself as a reverse forwarder candidate in the network. Then, the forwarder candidate updates its records and after setting the priority in the routing table and forwards the received message to its downstream link candidates located on the lower levels. The designed mechanism by diverting the incoming traffic to another suitable routing path provides load balancing when the offered data traffic load exceeds the available node’s buffer capacity in UWSNs. This avoids the data packets loss in UWSNs. During a new route discovery process, each iteration in the routing table is confirmed with their priorities so that no lower priority candidate is selected as the next hop relay node in the network. This entire mechanism guarantees transmission reliability even when a certain percentage of acoustic sensor nodes die in the network.

## 4. Performance Analysis

The following section discusses the channels models, simulation settings and results obtained during simulation studies.

### 4.1. Channel Model and Simulation Settings

In this study, we employ the widely used acoustic channel model explained in [41,42,43] to simulate the dynamics of acoustic communication in UWSNs. In this model, the path loss is mainly caused by the geometrical spreading and signal attenuation associated with frequency dependent absorption is calculated as:(35)$$10\log A\;\left(d,f\right)/A_{O}=k\times 10\log d+d\times 10\log a{\left(f\right)}^{d}\;$$

The absorption coefficient a(f) using the Thorp’s formula [44] is given as:(36)$$10\log a\left(f\right)=\frac{0.11\times f^{2}}{1+f^{2}}+\frac{44\times f^{2}}{4100+f^{2}}+2.75\times 10^{-4}f^{2}+0.003$$
in which, the factors transmission loss A(d,f) and noise level N(f) are functions of the distance (d) and frequency. It is worth mentioning that noise decreases with frequency and turbulence, shipping activities, breaking waves and thermals are the primary sources of ambient noise. Therefore, the noise that affects the underwater acoustic channel is originated from ambient and site-specific sources which is modelled using four sources, including waves, shipping, thermal noise and turbulence, respectively. In addition, the widely used general energy consumption model discussed in [45,46,47] is employed in the proposed scheme. In fact, we evaluate the performance of QoSRP against well-known LRP [24], QERP [26] and MERP [10] routing protocols developed for UWSNs-based events monitoring applications in UWEs. The network simulation tools, namely NS2 and AquaSim 2.0 are used to implement the schemes in random network topologies. During simulations, a set of 250 ASNs in a region of size length (m) × width (m) × depth (m) is randomly deployed to simulate the continuous events monitoring using acoustic sensor network in UWEs. The sea surface buoys is deployed in the middle of the UASNs. The initial energy of each ASN and sea surface sink were set to 100 J and 100 kJ with average ocean depth is 1 mile (≈1.6 km). The beam widths of each ASN is varied between 0 and 360 degrees in each set of experiments while the acoustic communication range of the sink was 200 m. The results of the simulations correspond to an average value of 53 runs. The settings of different parameters is given in Table 3.

### 4.2. Results and Discussion

In this section, the results obtained from the single channel protocols with interference are compared with the multichannel protocol with the same interference rate of 15% (low), 35% (moderate) and 55% (high) in UWSNs. During simulation studies, it is noticed that the packets delivery ratio is found high in all routing protocols when the interference rate level is 15%. The interference does not heavily affect the transmissions in a single channel protocol at the beginning as the interference is not frequent enough, showing the average 87%, 90%, 92% reception rates in LRP, QERP and MERP in the first 30 s as shown in Figure 6. This is because the nodes in these routing schemes have enough time to recover from the interference through retransmissions. However, the packets delivery ratio decreases rapidly over time when the interference level is linearly increasing from 15% to 55%. From low to high, the packet loss increases over time and always goes up until 16%, 14% and 11% in MERP, QERP and LRP routing protocols. This rate is found to be extremely low around 1% in QoSRP nearly throughout the simulation period, compared to all other routing protocols in UWSNs. However, the overall packets delivery ratio is recorded high around 89% in MERP compared to 86% and 84% in the QERP and LRP routing protocols in UWSNs. During simulations, it is observed that due to the presence of interference, the single-channel protocol shows deteriorate reception rate in UWSNs. The reason for these losses is because the network is being congested with retransmission packets. The congestion management profile of each routing protocol is shown in Figure 7. It indicates that the congestion management efficiency of QERP is higher around 90% compared to the MERP and LRP routing protocols. On the other hand, the congestion management profile of MERP and LRP are overlapping each other and is around 83% and 82%, respectively.

In these single-channel protocols, the node does not have enough time to recover from the high interference to retransmit and drops most of the packets. There are more packets being dropped over time and the node stops receiving packets as it does not have enough buffer space to store the incoming packet and the channel becomes congested. However, the MERP protocol data delivery performance improves as the interference rate increases since it has more time to recover compared to the QERP and LRP routing protocols. On the contrary, the overall packets delivery ratio is recorded high at around 99% in QoSRP in UWSNs. In QoSRP, the number of packets received at the sink is observed around 88% when the vacant channel’s record is found to be around 5 at each ASN in UWSNs. 

This low packet delivery ratio is due to the method utilizing low priority channels which cannot effectively mitigate the interference and noise effects for packets transmission in UWSNs. This level rapidly increases up to 96% when the ASNs keeps the record around 9 vacant channels in UWSNs. As soon as the packets reception rate is increasing, the ASN starts to block the channels that are facing high interference and noise issues for a predefined amount of time by assigning a low priority level in the routing table. By doing this, it ensures that the rest of the node’s listening channels are good channels. This enables the use of all other available channels without blacklisting any channel until it is sure that it is a bad channel through the negotiation process. Therefore, the QoSRP periodically orders the channel based on their priority each time before making a decision on the new channel in order to avoid the interference channel. Thus, it enhances the ability of an acoustic node to find a vacant channel in a robust manner with higher data rates at low interference, which effectively increase the network throughput in QoSRP around 96% compared to other routing protocols as shown in Figure 8. This is because as there are fewer channels stored in the routing table with high priority and no interference, some of the nodes are using the channels with extremely low interference in the same or different regions in UWSNs. The nodes could recover from low interference and retransmit packets as the other nodes are on the other channels, thus the nodes do not compete for transmissions on the channel. Therefore, there is a variation of the packet reception rate, which increased over time in UWSNs. Finally, the packets delivery ratio (PDR) rate is recorded around 99% when the maximum vacant channels record up to 11 at each ASN in UWSNs. The key reason for this high PDR is, the secondary users’ activities is higher during packets forwarding compared to the primary user activities for a long time in UWSNs. On the other hand, this high PDR is also due to utilizing the most appropriate channel to mitigate the interference and noise effects during packets transmission in UWSNs. In addition, the other main reason for high PDR and throughput is the efficient performance of the channel detection algorithm in UWSNs. The channel detecting probability is noticed to be high with low FA𝓈 and MeD probabilities as shown in Figure 9; Figure 10, respectively. Thus, in all low, intermediate and high interference cases, the channel detection probability is high for QoSRP at around 87% to 91% in both individual and group sensing cases, even though there is interference in the network. In case of high interferences, the QoSRP selects certain channels to change into after checking the channels’ condition and hops to another channel quickly by efficiently detecting the primary user signals to keep the error and loss rates to a minimum as shown in Figure 11. This indicates that the packet error rate in all routing schemes increases with the increasing node density. The average packet error rate of LPR is observed to be high around 10% compared to QERP and MERP recorded up to 7% and 8% in UWSNs. Usually, QERP and MERP are overlapping each other’s and try to achieve a minimum packet error rate in UWSNs. On the contrary, the packet error rate is noticed to be around 2%–3% in the QoSRP routing protocol in UWSNs.

Consequently, each node has different listening and transmitting channels. When the node is awake, it waits for the incoming packets on its listening channel. If the node has a packet to send, it will switch to the desired capacity-aware channel based on the channel information from the neighboring table. The channel quality table is built over time at each hop helps to learn good and bad channels based on the previous history information of message exchange processes. The channel switching takes at most a few milliseconds to switch to the transmission channel. The QoSRP uses a transmission phase-lock that is based on the previous history wake up or sleep activity information of the primary users and, therefore, the transmission node knows the receiver wake up phase. The node starts transmitting just before the receiver is expected to be awake. The channel switching happens shortly before the receiver is ready to receive the packets, thus the time taken in channel switching does not affect the packet reception in the network. The node goes back to sleep once the transmission has succeeded or reached the maximum number of retransmissions. In the next iteration, the node is reset and wakes up on its listening channel. The channel reset is done in these cases: (i) the queue buffer is empty, (ii) before sending the next packet from the queue buffer, and (iii) the last packet in the queue buffer has been sent. This reset is done to avoid any delay in packet reception that could happen when the node is awake. In load-balanced routing, the workload is evenly distributed in the network, which as a result, distributes the energy consumption across the nodes. The static channel-based studies, however, do not consider the appropriate energy or the battery level as the performance metric but instead use the node’s workload value. Therefore, they perform poorly in load balancing in the network. This rate is found more in MERP and LRP compared to the QERP routing protocol. On the other hand, the QoSRP uses routing through the good neighborhood, which provides alternative routes instead of concentrating on a single good path to ensure that the workload is widely spread and no specific nodes are being used excessively.

The neighborhood metric uses the information regarding the quality of the surrounding neighborhood, which are the forwarding path value and the neighborhood influence on the node in making a decision based on the priority. If the current path becomes congested or unavailable, it switches to the alternative route, which is through a node closer to the destination to improve the overall throughput and data rates in the network. Thus, the next hop neighbor that is not selected as the forwarder becomes the alternative route if the current path is unavailable. The neighborhood metrics allow a set of forwarding routes to be used to enable network load distribution, which as a result, helps to improve the network load balancing and reduces the energy consumption, and latency. All the above factors lead to low latency in QoSRP compared to QERP, MERP and LRP routing schemes as shown in Figure 12. The overall delay performance in QoSRP routing protocol with 95% confidence intervals is recorded up to 700 ms. By contrast, the latency performance of QERP is observed well around 800 ms compared to MERP and LRP recorded around 890 and 1000 ms in UWSNs. The low latency of QERP is due to it finding alternative routes in a robust manner. On the contrary, the latency performance of MERP is better due to it spending less time handling congestion issues in the packets forwarding process. The LRP scheme performs insignificant than all routing schemes due to spending a prominent time during routes finding process in the network. In sum, the proposed scheme is extremely for the underwater monitoring applications that requires high data rates, throughput with low latency, packet error rate and network energy-consumption performance.

## 5. Conclusions

The design of a QoS-aware routing protocol for time-critical events data gathering is the major concern in UWSNs. Therefore, this paper proposed a new cross-layer QoS-aware multichannel data-gathering protocol for UWSNs-based time-critical marine monitoring applications. The proposed scheme takes into account the time and location-dependent acoustic channel communication characteristics during time-critical events data gathering and relaying towards the sink. In the proposed QoSRP scheme, the UWCD mechanism successfully detects vacant channels for ASNs with a high probability of detection and low probability of missed-detection and FA𝓈 in the highly dynamic network topology. On the other hand, the UWCA mechanism successfully assigns high data rates channels with longer idle probability in a robust manner for SASNs in UWEs. Finally, the UWPF mechanism exploits the relay ASNs by employing hybrid angle-based and greedy routing mechanisms to their NASNs to route packets around connectivity voids and shadow zones in UWSNs. Thus, the packets forwarding by employing simple topology information (hop count), angle information, residual energy and previous history of successful transmissions in a greedy manner towards the sink greatly reduce the congestion, data path loops, and balances the energy consumption load of UWSNs. Our results revealed that the proposed QoSRP scheme performs better compared to existing routing schemes in terms of data delivery rates, packet error rates, throughput, latency, congestion, and data traffic load balancing with the expense of communication overheads in UWSNs. As future work, the researchers aim to focus on the ASNs’ mobility and energy consumption issues with different QoS requirements in order to achieve a more realistic approach for time-critical applications of UWSNs.

## Figures and Tables

**Figure 1 sensors-19-04762-f001:**
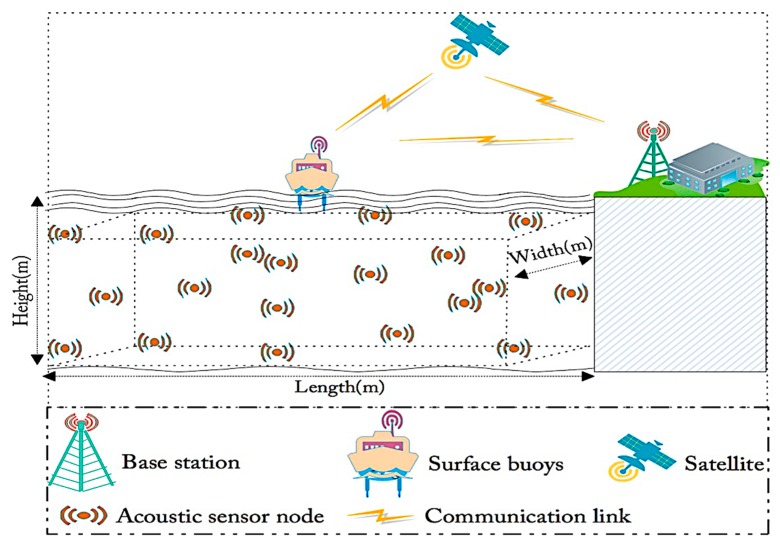
Network model in quality of service routing protocol (QoSRP) protocol.

**Figure 2 sensors-19-04762-f002:**
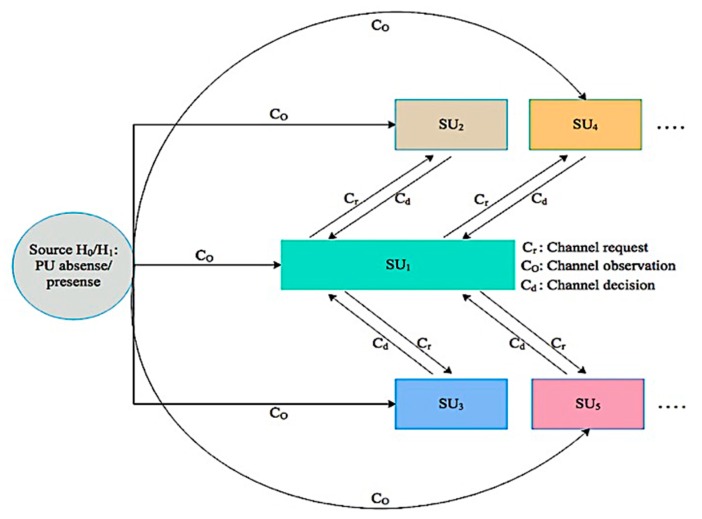
The channel request, monitoring and decision process.

**Figure 3 sensors-19-04762-f003:**
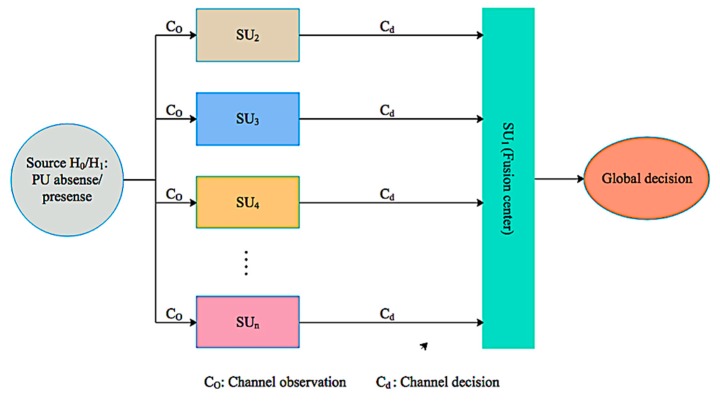
The parallel sensing mechanism.

**Figure 4 sensors-19-04762-f004:**
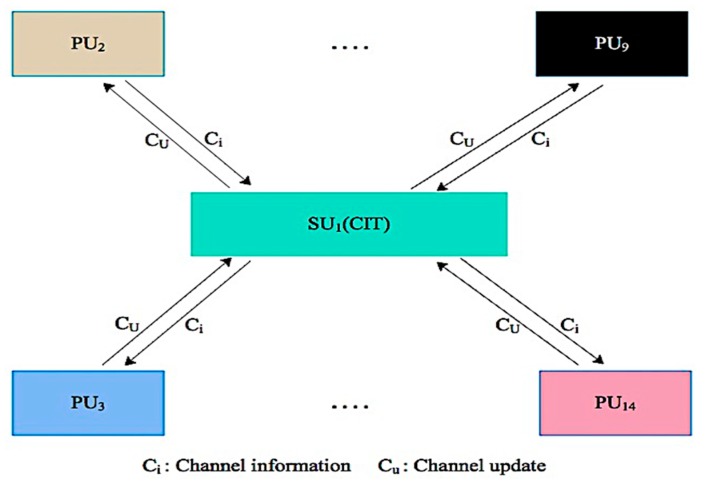
Channel information updating process.

**Figure 5 sensors-19-04762-f005:**
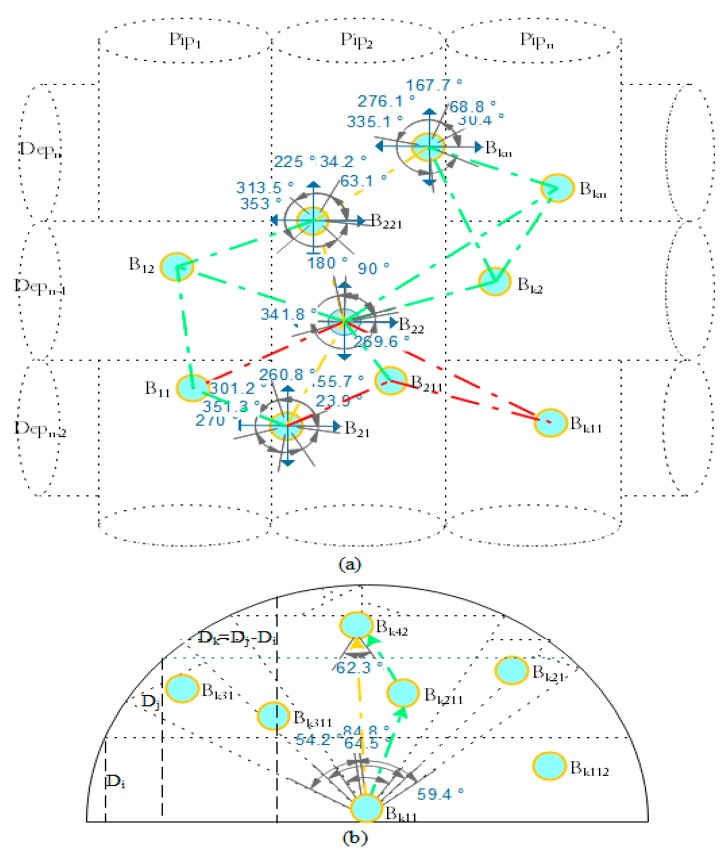
Packets forwarding mechanism in QoSRP. **(a**) relaying data; (**b**) angle formation.

**Figure 6 sensors-19-04762-f006:**
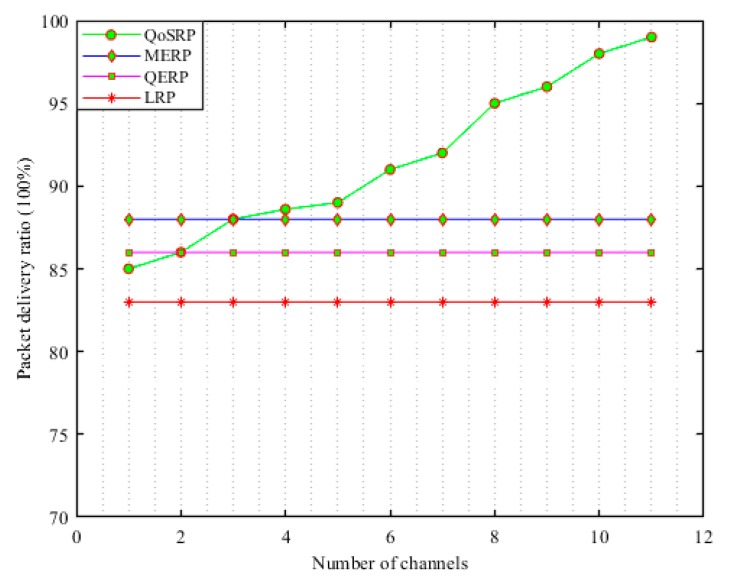
Packets delivery ratio vs. number of channels.

**Figure 7 sensors-19-04762-f007:**
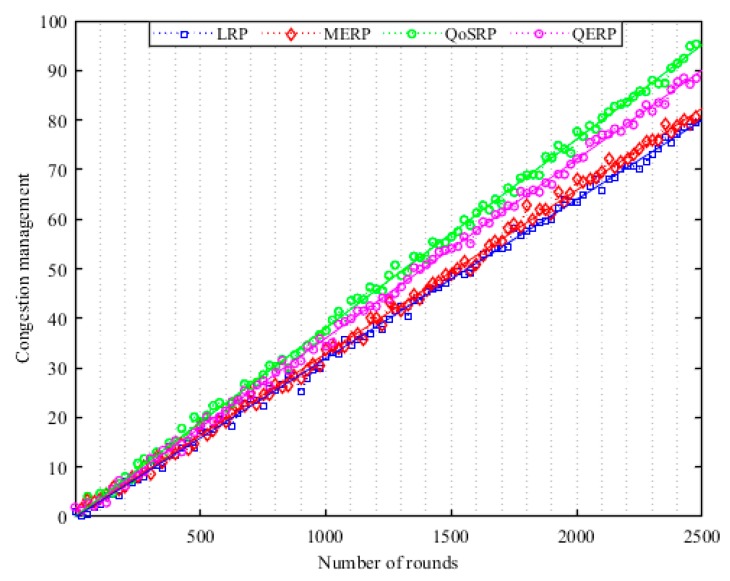
Congestion management vs. number of rounds.

**Figure 8 sensors-19-04762-f008:**
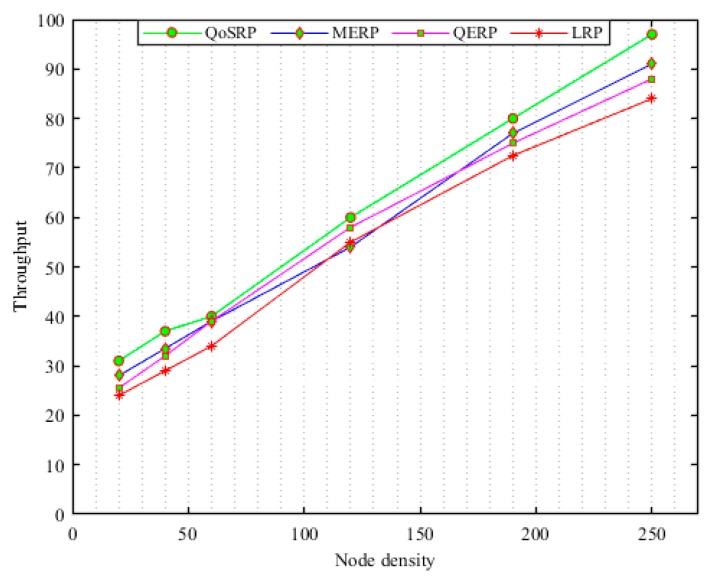
Throughput vs. number of acoustic nodes.

**Figure 9 sensors-19-04762-f009:**
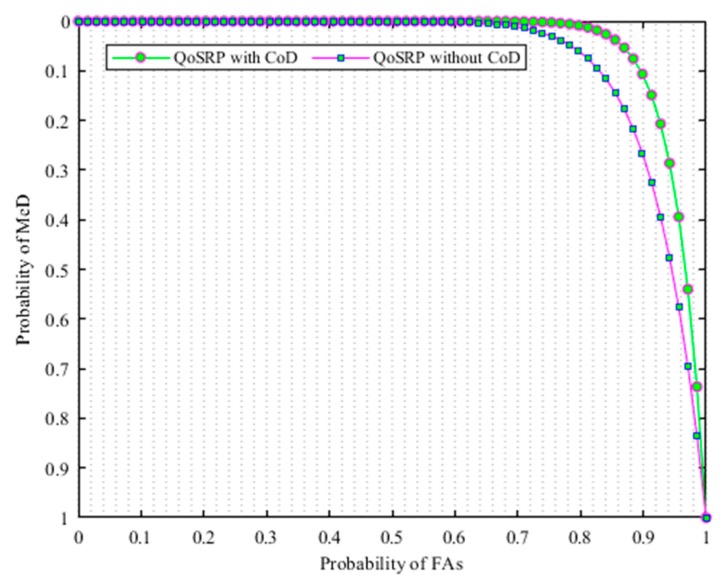
Probability of missed detection vs. false alarms.

**Figure 10 sensors-19-04762-f010:**
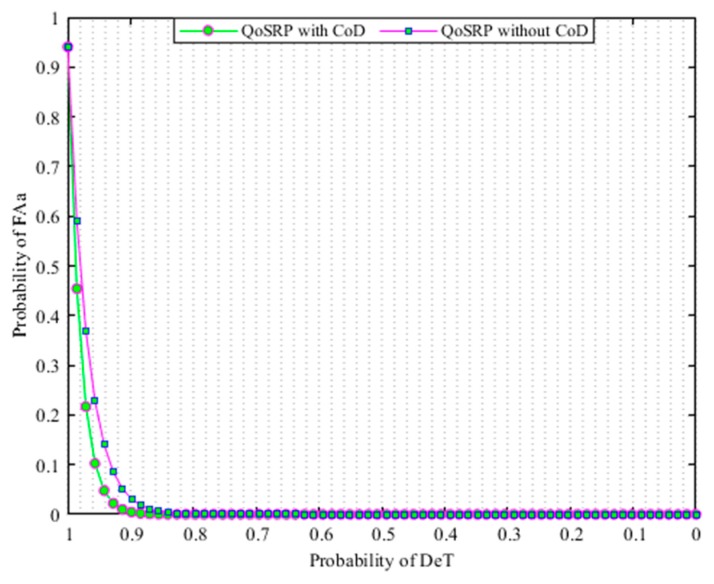
Probability of false alarms vs. detection.

**Figure 11 sensors-19-04762-f011:**
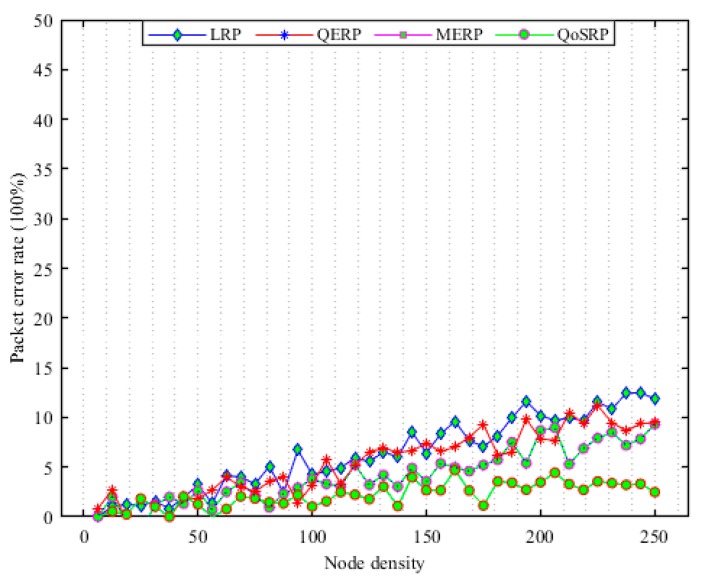
Packets error rate vs. number of acoustic nodes.

**Figure 12 sensors-19-04762-f012:**
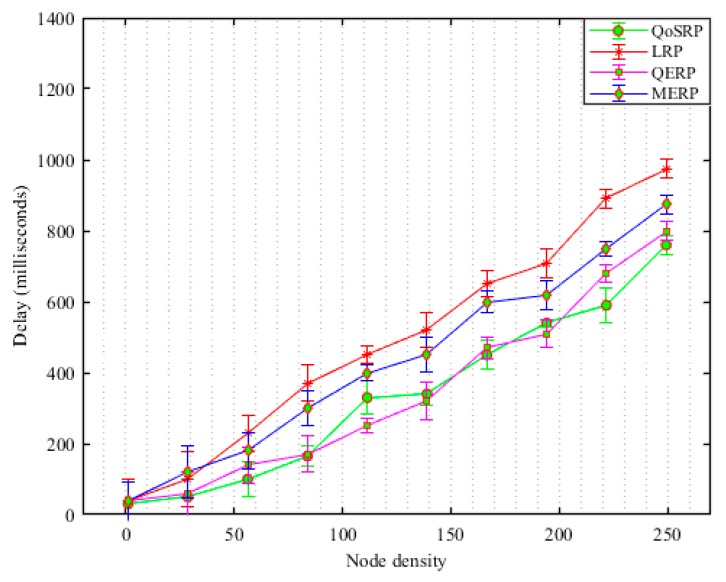
Delay vs. number of acoustic nodes.

**Table 1 sensors-19-04762-t001:** Comparison of routing schemes in underwater wireless sensor networks (UWSNs).

SrNo.	Routing Schemes	Multi-Channel	Static-Channel	Architecture	Channel Sensing	Dynamic Channl Assign	Channel Capacity	Packet Delivery Ratio	Delay	Energy Consumption	Throughput	Congestion	Packet Error Rates
1	MCUW [17]		✔	Flat				✔		✔			
2	DSDBR [12]		✔	Flat				✔	✔	✔			
3	AEDG [18]		✔	Tree				✔	✔		✔		
4	CARP [19]		✔	Flat				✔	✔		✔		
5	EGRCs [23]		✔	Clustering				✔	✔	✔			
6	GEDAR [13]		✔	Flat				✔	✔	✔			
7	HydroCast [15]		✔	Flat				✔	✔	✔			
8	LRP [24]		✔	Clustering				✔	✔	✔	✔		
9	ENMR [28]		✔	Flat				✔	✔	✔			
10	AREP [29]		✔	Flat				✔	✔	✔			
11	E-CBCCP [25]		✔	Clustering				✔		✔			
12	EDOVE [14]		✔	Flat				✔	✔	✔			
13	QERP [26]		✔	Clustering				✔	✔	✔			
14	RE-PBR [2]		✔	Flat				✔	✔	✔			
15	MRP [30]		✔	Flat				✔	✔	✔			
16	SDCS [27]		✔	Hybrid				✔	✔	✔			
17	RECRP [20]		✔	Flat				✔	✔	✔			
18	CACR [21]		✔	Flat				✔	✔	✔			
19	DVOR [16]		✔	Flat				✔	✔				
20	DQELR [22]		✔	Flat				✔	✔	✔	✔		
21	RACAA [31]		✔	Flat				✔	✔	✔			
22	MERP [10]		✔	Clustering				✔	✔	✔		✔	
23	QoSRP	✔		Flat	✔	✔	✔	✔	✔		✔	✔	✔

**Table 2 sensors-19-04762-t002:** Notations used in QoSRP.

Notation	Description
$C_{𝓊}$	is the utilization of a channel $C_{𝒾}$ in UWSNs
$T_{𝓅}$	is the transmission power of a channel $C_{𝒾}$ in UWSNs
$I_{𝓃𝓉𝒻}$	is the interference in UWSNs
$P_{ℯ𝓇}$	is the packet error rate in UWSNs
$C_{𝓉}$	is the channel management cost
$T_{𝓅\left(𝓂𝒶𝓍\right)}\left(C_{𝒾}\right)$	is the maximum transmission power for a channel $C_{𝒾}$ in UWSNs
$T_{𝓅\left(𝓂𝒾𝓃\right)}\left(C_{𝒾}\right)$	is the minimum transmission power for a channel $C_{𝒾}$ in UWSNs
$T_{𝓅\left(𝓉𝒽𝓇ℯ𝓈𝒽𝑜\ell 𝒹\right)}$	is the threshold value for transmission powerin UWSNs
$G_{\left(SU_{\;𝒿},\;PS_{𝒿}\right)}$	is the gain for a channel $C_{𝒾}$ from $SU_{\;𝒿}$ to $PS_{𝒿}$ in the network
$R_{I_{ntf}}\left(SU_{\;𝒿}\right)$	Is the interference range of a secondary user $SU_{\;𝒿}$ in the network
$P_{\ell 𝑜𝓈𝓈}\left({𝒹}_{𝓂𝒶𝓍}\right)$	is the path loss over a maximum distance between a transmitter and receiver (i.e., coverage radius)
$P_{\ell 𝑜𝓈𝓈}\left(𝒹\right)$	is the total path loss at a distance d from the transmitter measured using log-normal shadowing model
$B_{I_{𝓃𝓉𝒻}\;𝓅\left(\left(𝓂𝒶𝓍\right)\right)}\left(\;C_{𝒾}\right)$	is the maximum background interference power at the receiver over a channel $C_{𝒾}$ in the network
$I_{𝓃𝓉𝒻}\left(𝒸𝑜𝓃𝓈\right)$	is the interference constraint in the UWSNs
$T_{𝒽𝓅}\left(C_{𝒾}\right)$	is the throughput of a channel 𝒾 in the network
σ	is the constant factors with a value less than 1 and greater than 0
$U_{𝓉}$	is the utility function with maximum value 1
$N_{𝑜}$	is the noise factor with maximum value 1 and minimum value 0
$PS_{𝒿}^{𝒽\left(𝓀\right)}$	is the number of channels stored by a $PU_{𝒾}$ or $SU_{\;𝒿}$ in time $\tau_{𝒾}$ with decreasing priority in the channel table
$\rho_{available}^{C_{𝒾}}\left(SU_{\;𝒿}\right)$	is the probability of a secondary user 𝒿 for the channel $C_{𝒾}$ in the network
$\rho_{\tau_{𝑜𝓃\to 𝑜ff}}^{C_{𝒾}}\left(PU_{𝒾}\right)$	is on and off the probability of a primary user 𝒾 on a channel $C_{𝒾}$ in the network
$\rho \;SU_{𝒿}\left(C_{𝒾}\right)$	is the collision probability ρ of a secondary user 𝒿 for the channel $C_{𝒾}$ in the network
$\rho \;PU_{𝒾}\left(C_{𝒾}\right)$	is the collision probability ρ of a primary user 𝒾 for the channel $C_{𝒾}$ in the network

**Table 3 sensors-19-04762-t003:** Values of parameters used in QoSRP.

Parameters	Value (s)
Channel	Underwater channel
Network topology	Random
Deployment area	1000 ×1000 ×500 $m^{3}$
Initial node energy	100 J
Initial sink energy	100 kJ
Number of nodes	250
Cost of high transmission	1.5 W
Cost of low transmission	1 W
Cost of reception	0.75 W
Idle power	0.01 W
Data aggregation power	0.35 W
Communication range of ASN	150 m
Acoustic transmission range of sink	200 m
Spreading values	1.5
Frequency	30.5 kHz
Number of Channels	11 (30.511, 30.518, 30.525, 30.532, 30.539, 30.546, 30.560, 30.553, 30.567, 30.574, 30.581) kHz
Maximum Bandwidth	30 kbps
Packet size	50 bytes
Control packet size	5 bytes
Packet generation rate	0.01~0.1 packets/s
Memory size	10 MB
Maximum sink and ASN	1 km
Antenna	Omni-directional
Simulation time per epoch	150 s
Number of runs	53

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
