# Peer review of "QoSRP: A Cross-Layer QoS Channel-Aware Routing Protocol for the Internet of Underwater Acoustic Sensor Networks"

_sensors, 2019, doi:10.3390/s19214762_

Round 1

Reviewer 1 Report

This paper proposes a QoS-based routing protocol in UASN,which exploits multi-channel cognitive technology to guarantee high probability of detection and low probability of missed detection and false alarms.The authors provide us a detailed review on related works and technology foundation,but there are still some problems,as follows.
1.Confusing notations and variables,which may perplex readers,e.g. formula (1) originally shows a multiple objectives optimization, and others equations including ∈.
2.There are some clerical error,such as ''can be simply can be formulated'' in row 439 and double chapter 3.3
3.Move Table 2 up as much as possible in order for easily understanding
4.How to achieve the factors listed in Eq.19?
5.In 937,how could an ASN know its distance to the sink?
6.I extremely doubt about the parameters listed in Table 3,different channel is guarded by a small 7 Hz frequency distance with unknown bandwidth,please clear why do you choose the parameters look like unresonablely.

Author Response

Responses to Reviewer 1 for SENSORS

Manuscript Number: Sensor-603975

QoSRP: A Cross-layer QoS Channel-aware Routing Protocol for the
Internet of Underwater Acoustic Sensor Networks

We thank the reviewer and the Editor for their constructive comments and valuable time in reviewing our paper. In this revised version, we listed all the comments that helped us to significantly improve our work, and the corresponding responses and revisions. We have updated the paper to incorporate the reviewer's comments and highlighted our changes in the revised text.

Reviewer #1 Comments:

        1. This paper proposes a QoS-based routing protocol in UASN,which exploits multi-channel cognitive technology to guarantee high probability of detection and low probability of missed detection and false alarms.

RESPONSE: We thank the reviewer for clearly understanding the entire working principle of our developed scheme and pointing out the importance of our research.

         2. Confusing notations and variables, which may perplex readers,e.g. formula (1) originally shows a multiple objectives optimization, and others equations including . 

RESPONSE: Thank you for your valuable comments. Based on the reviewer’s comments, we have double-checked the entire notations and variables used in the revised text. In addition, as discussed in Section 2 in detail, we employ MILP programming to formulate the problems. Therefore, it is essential to describe every single objective separately in Eq.1 and use and other notations to indicate the relationship between different variables. In fact, these notations show the impact on each Eq. differently by following the key objective function as defined in Eq.1.

        3. There are some clerical error,such as ''can be simply can be formulated'' in row 439 and double chapter 3.3

RESPONSE: We are grateful to the reviewer for noticing the grammatical and typo errors. Based on the reviewer’s constructive comments, the grammatical errors are now fixed and highlighted with red in the revised paper. In addition, we have fixed the issue of double chapter numbering in the revised text.

         4. Move Table 2 up as much as possible in order for easily understanding

RESPONSE: Really, thank you for your encouraging comments and fruitful suggestions. Based on the reviewer’s comment, Table 2 has been shifted up in the beginning of Section 3 in the revised text. Now, it is more clear for the readers to understand the symbols before look insight the text.

         5. How to achieve the factors listed in Eq.19?

RESPONSE: Thanks for your comments. In fact, these are basic acoustic channel modeling factors that can be found almost in recent studies. Therefore, we have used these factors in our scheme during modeling acoustic communication for UWSNs. In this respect, some important reference papers are listed below:

Javaid et al., "Delay-sensitive routing schemes for underwater acoustic sensor networks," International Journal of Distributed Sensor Networks, vol. 2015, p. 1, 2015. W. Coutinho, A. Boukerche, L. F. Vieira, and A. A. Loureiro, "Geographic and Opportunistic Routing for Underwater Sensor Networks," IEEE Transactions on Computers, vol. 65, no. 2, pp. 548-561, 2016. Bouk, S. Ahmed, K.-J. Park, and Y. Eun, "Edove: Energy and depth variance-based opportunistic void avoidance scheme for underwater acoustic sensor networks," Sensors, vol. 17, no. 10, p. 2212, 2017. Noh et al., "Hydrocast: Pressure routing for underwater sensor networks," IEEE Transactions on Vehicular Technology, vol. 65, no. 1, pp. 333-347, 2016. Guan, F. Ji, Y. Liu, H. Yu, and W. Chen, "Distance-Vector based Opportunistic Routing for Underwater Acoustic Sensor Networks," IEEE Internet of Things Journal, 2019. Su, R. Venkatesan, and C. Li, "An energy-efficient relay node selection scheme for underwater acoustic sensor networks," Cyber-Physical Systems, vol. 1, no. 2-4, pp. 160-179, 2015. Ilyas et al., "AEDG: AUV-aided efficient data gathering routing protocol for underwater wireless sensor networks," Procedia Computer Science, vol. 52, pp. 568-575, 2015. Basagni, C. Petrioli, R. Petroccia, and D. Spaccini, "CARP: A channel-aware routing protocol for underwater acoustic wireless networks," Ad Hoc Networks, vol. 34, pp. 92-104, 2016. Liu, M. Yu, X. Wang, Y. Liu, X. Wei, and J. Cui, "RECRP: An Underwater Reliable Energy-Efficient Cross-Layer Routing Protocol," Sensors, vol. 18, no. 12, p. 4148, 2018.

         6. In 937,how could an ASN know its distance to the sink? 

RESPONSE: Thank you again for your valuable comments.  In fact, we have used the localization scheme discussed in reference [32] in Section 3.1 network model in order to identify the neighboring nodes as well as the sink. In addition, the neighboring and sink discovery process has been explained in detail in subsection 3.4.1 and 3.4.2, respectively.

           7. I extremely doubt about the parameters listed in Table 3,different channel is guarded by a small 7 Hz frequency distance with unknown bandwidth,please clear why do you choose the parameters look like unresonablely.

RESPONSE: We really appreciate the reviewer’s comments. Dear Professor, the parameters and their values used in our simulation are common which can be found in the recent studies. In addition, one of the key aims of the research project and funding is to explore the narrow bandwidth channels for efficient and reliable communication in UWSNs. However, we totally agree with you, there exist several other frequency bands which can be explored for higher data rates in UWSNs. Consequently, the common underwater parameters and their values can be found in the following links.

Wang, H. Gao, X. Xu, J. Jiang, and D. Yue, "An energy-efficient reliable data transmission scheme for complex environmental monitoring in underwater acoustic sensor networks," IEEE Sensors Journal, vol. 16, no. 11, pp. 4051-4062, 2016. Su, R. Fan, X. Fu, and Z. Jin, "DQELR: An Adaptive Deep Q-Network-Based Energy-and Latency-Aware Routing Protocol Design for Underwater Acoustic Sensor Networks," IEEE Access, vol. 7, pp. 9091-9104, 2019. Noh et al., "Hydrocast: Pressure routing for underwater sensor networks," IEEE Transactions on Vehicular Technology, vol. 65, no. 1, pp. 333-347, 2016. Basagni, C. Petrioli, R. Petroccia, and D. Spaccini, "CARP: A channel-aware routing protocol for underwater acoustic wireless networks," Ad Hoc Networks, vol. 34, pp. 92-104, 2016. Han, S. Shen, H. Song, T. Yang, and W. Zhang, "A stratification-based data collection scheme in underwater acoustic sensor networks," IEEE Transactions on Vehicular Technology, vol. 67, no. 11, pp. 10671-10682, 2018. Yuan, C. Liang, M. Kaneko, X. Chen, and D. Hogrefe, "Topology control for energy-efficient localization in mobile underwater sensor networks using Stackelberg game," IEEE Transactions on Vehicular Technology, vol. 68, no. 2, pp. 1487-1500, 2018. Gomathi and J. M. L. Manickam, "Energy efficient shortest path routing protocol for underwater acoustic wireless sensor network," Wireless Personal Communications, vol. 98, no. 1, pp. 843-856, 2018. Rani, S. H. Ahmed, J. Malhotra, and R. Talwar, "Energy efficient chain based routing protocol for underwater wireless sensor networks," Journal of Network and Computer Applications, vol. 92, pp. 42-50, 2017.

Reviewer 2 Report

Please reduce the number of words to make the paper more concise. Too many sentences are used to illustrate the reasons to choose the acoustic communication in Section 1. In section 1, the paper says the large-scale UASNs are expensive and limited to experimental settings, but this problem is not solved by QoSRP. The contributions of your proposed works should be listed after the introduction of the existing studies and challenges. Introduction of existing studies should not be set as a section. So, the structure of the paper should be revised. The introduction of UWCD algorithm is tedious, the flow chart or diagram would make the illustration more concise. The length of section 4 is too short. If no more contents in section 4, please add it to section 5. In section 5, only simulation analyses are included. Field research should be included in this paper to prove the validity of QoSRP.

Author Response

Responses to Reviewer 2 for SENSORS

Manuscript Number: Sensor-603975

QoSRP: A Cross-layer QoS Channel-aware Routing Protocol for the
Internet of Underwater Acoustic Sensor Networks

We thank the reviewer and the Editor for their constructive comments and valuable time in reviewing our paper. In this revised version, we listed all the comments that helped us to significantly improve our work, and the corresponding responses and revisions. We have updated the paper to incorporate the reviewer comments and highlighted our changes in the revised text.

Reviewer #1 Comments:

Please reduce the number of words to make the paper more concise. Too many sentences are used to illustrate the reasons to choose the acoustic communication in Section 1. In section 1, the paper says the large-scale UASNs are expensive and limited to experimental settings, but this problem is not solved by QoSRP. 

RESPONSE: We really appreciate the reviewer’s encouraging comments and fruitful suggestions. Based on the reviewer’s comments, we have revised the Section-1 and removed the misleading term“large-scale” in the text.

      2. The contributions of your proposed works should be listed after the                        introduction of the existing studies and challenges.

RESPONSE: We are grateful to the reviewer’s comments. Based on the reviewer’s comments, we have listed our contributions after introduction of the existing studies and challenges in the revised text.

      3. Introduction of existing studies should not be set as a section. So, the                    structure of the paper should be revised.

RESPONSE: We thank the reviewer’s comments. In fact, this is one of the basic paper formatting requirements of the SENSORS journal. Therefore, we have separated Section 2 from Section 1 in the text.

      4. The introduction of UWCD algorithm is tedious, the flow chart or                            diagram would make the illustration more concise.

RESPONSE: Once again, thank you for your constructive comments. Dear Professor, due to restricted manuscript page length limitations and given time, further detail of the study cannot be extended in the revised paper. However, based on the reviewer’s constructive comments, we have clearly explained the UWCD algorithm in the revised text.

     5. The length of section 4 is too short. If no more contents in section 4, please           add it to section 5. In section 5, only simulation analyses are included. Field           research should be included in this paper to prove the validity of QoSRP.

RESPONSE: Again, we are grateful to the reviewer’s constructive comments. Based on the reviewer’s comments, we have combined Section 4 and Section 5 in the revised text. Now, it is more clear, concise and attractive to the readers.

Reviewer 3 Report

This paper with the title QoSRP: A Cross-layer QoS Channel-aware Routing  Protocol for the Internet of Underwater Acoustic Sensor Networks, proposes a novel cross-layer QoS-aware multichannel routing protocol called QoSRP for the internet of UWSNs-based time-critical marine monitoring applications. After a deeply study and revision of state of art in the field the authors describing an original protocol. Reveal by simulation that the QoSRP protocol improve for example the throughput, diminish the error packets and delay when the number of nodes increase and the congestion management efficiency is better compared to existing routing schemes in UWSNs ( QERP, MERP,  LRP).

I think that this protocol is scientifically important to the field and describes several important innovations in UWSN design. The work is well-organized and generally well-written. I recommend that it be published, after minor revisions.

The following changes should be introduced in the document:

In line 131 change "with the expense" by  "at the expense of"

In figure 1 only two dimensions are shown, width and depth add the third dimension length

In line 354: What mean PU signal ? the first time it appears in the text its meaning must be described. Do you mean PU primary users?

In line 396:  What mean SU ? the same comment that in line 354. Do you mean secondary users?.

In line 465 change
for a particular time while the constaints in Eq. (5e) satisfy the satement.
by
for a particular time while the constraints in Eq. (5e) satisfy the statement.

In last line of table 2 change secondary user by primary user

In line 1025 appear neighbors, why use here American English when in rest of text use British English neighbours?.

In table 3 appear different units in the SI J,kHz,W and its name complete for example, Watts (W), kilohertz (kHz). I think redundancy  its not necessary. Write only W or Watts and the same for the others units. Moreover the units for bandwidth are wrong Kbps replace by kbps (k in lowercase, K is the fundamental unit in the SI Kelvin).

Finally I suggest to build a QoSRP (in a real experimental setup on a reduced scale) to verify this important advance that present this new proposed protocol.

Author Response

Responses to Reviewer 3 for SENSORS

Manuscript Number: Sensor-603975

QoSRP: A Cross-layer QoS Channel-aware Routing Protocol for the
Internet of Underwater Acoustic Sensor Networks

We thank the reviewer and the Editor for their constructive comments and valuable time in reviewing our paper. In this revised version, we listed all the comments that helped us to significantly improve our work, and the corresponding responses and revisions. We have updated the paper to incorporate the reviewer's comments and highlighted our changes in the revised text.

Reviewer #3 Comments:

1. This paper with the title QoSRP: A Cross-layer QoS Channel-aware Routing Protocol for the Internet of Underwater Acoustic Sensor Networks, proposes a novel cross-layer QoS-aware multichannel routing protocol called QoSRP for the internet of UWSNs-based time-critical marine monitoring applications. After a deeply study and revision of state of art in the field the authors describing an original protocol. Reveal by simulation that the QoSRP protocol improve for example the throughput, diminish the error packets and delay when the number of nodes increase and the congestion management efficiency is better compared to existing routing schemes in UWSNs ( QERP, MERP, LRP). I think that this protocol is scientifically important to the field and describes several important innovations in UWSN design. The work is well-organized and generally well-written.

RESPONSE: We really appreciate the reviewer’s encouraging comments. We thank the reviewer for clearly understanding the entire working principle of our developed scheme and pointing out the importance of our research.

2. In line 131 change "with the expense" by "at the expense of"

RESPONSE: Thank you for your valuable comments. Based on the reviewer’s comment, the required correction has been made and highlighted with red in the revised text.

3. In figure 1 only two dimensions are shown, width and depth add the third dimension length 

RESPONSE: Thank you again for your valuable comments. Based on the reviewer’s comment, we have added three-dimensional ‘figure 1’ in subsection 3.1.

4. In line 354: What mean PU signal ? the first time it appears in the text its meaning must be described. Do you mean PU primary users? 

RESPONSE: We really appreciate the reviewer’s comments. Based on the reviewer’s comment, the missing terms have been defined throughout the paper. Now, it is more clear for the readers.

5. In line 396: What mean SU ? the same comment that in line 354. Do you mean secondary users?.

RESPONSE: We really appreciate the reviewer’s encouraging comments. Based on the reviewer’s comment, we have defined the missing terms and highlighted with red in the revised text.

6. In line 465 change for a particular time while the constaints in Eq. (5e) satisfy the satement  by  for a particular time while the constraints in Eq. (5e) satisfy the statement.

RESPONSE: Really, thank you for your encouraging comments and fruitful suggestions. Based on the reviewer’s comment, we have changed the sentence in the revised text.

7. In last line of table 2 change secondary user by primary user

RESPONSE: We really appreciate the reviewer’s encouraging comments. Based on the reviewer’s comment, the required correction has been made and highlighted with red in the revised text.

8. In line 1025 appear neighbors, why use here American English when in rest of text use British English neighbours?.

RESPONSE: We are grateful to the reviewer for noticing the typo and grammatical errors. The typo and grammatical errors are now fixed in the revised paper.

9. In table 3 appear different units in the SI J,kHz,W and its name complete for example, Watts (W), kilohertz (kHz). I think redundancy its not necessary. Write only W or Watts and the same for the others units. Moreover the units for bandwidth are wrong Kbps replace by kbps (k in lowercase, K is the fundamental unit in the SI Kelvin).

RESPONSE: We are grateful to the reviewer for noticing different units in Table 3. Based on the reviewer’s comment, the required correction has been made and highlighted with red in the revised text.

10. Finally I suggest to build a QoSRP (in a real experimental setup on a reduced scale) to verify this important advance that present this new proposed protocol.

RESPONSE: We are grateful to the reviewer’s constructive comments. Dear Professor, due to given time and budget, the further detail of simulation analysis cannot be extended in the revised paper. However, we found your comments valuable and therefore will consider in the future.
